# IBMA: Information Bottleneck-Based Multimodal Alignment

**Yancheng Wang** [1]  **Zeyu Dong** [1]  **Dongfang Sun** [1]  **Alvin C. Silva** [2]  **Teresa Wu** [1]  **Yingzhen Yang** [1]

## Abstract

Multimodal learning aims to integrate information from heterogeneous data sources to improve representation quality and downstream task performance. A key challenge lies in aligning modality-specific representations while suppressing modality-dependent noise and redundancy. The Information Bottleneck (IB) principle provides a principled framework for learning task-relevant representations. Existing multimodal IB methods primarily apply the IB principle to fused multimodal representation and rely on restrictive distributional assumptions, such as Gaussian latent priors induced by variational autoencoders, which may not hold in practice. In this paper, we propose Information Bottleneck–based Multimodal Alignment (IBMA), a novel multimodal learning framework that enforces the IB principle for both the fused multimodal representation and modality-specific representations. IBMA introduces modality-specific representation alignment that guides each modality-specific encoder to learn informative and task-relevant representations aligned with the complementary modality, thereby enhancing cross-modal semantic consistency. Moreover, we derive a novel, efficient, and distribution-free variational upper bound for the IB loss that avoids unrealistic assumptions on latent feature distributions and is readily optimized using standard stochastic gradient descent. Extensive experiments demonstrate that IBMA achieves superior performance compared to existing multimodal learning methods, validating the effectiveness of modality-specific representation alignment. The code for IBMA is available at https://github.com/

Statistical-Deep-Learning/IBMA.

## 1. Introduction

Multimodal learning integrates heterogeneous data sources such as images and text to learn robust representations for downstream tasks (Li et al., 2024; Sleeman et al., 2023; Yuan et al., 2025). Most existing approaches employ separate modality-specific encoders and fuse them via cross-modal attention (Cai et al., 2023; Lu et al., 2023; Wu et al., 2025) or graph neural networks (Liu et al., 2025; Zhang et al., 2023). However, multimodal learning remains challenging due to modality-specific noise, redundancy, and semantic misalignment (Mai et al., 2023a; Wu et al., 2025). While alignment methods such as contrastive learning (Boecking et al., 2022; Guo et al., 2024; Xie et al., 2025; Zhang et al., 2024a; Zou et al., 2023) promote cross-modal consistency, they do not explicitly suppress redundant or noisy information within individual modalities. To explicitly control the trade-off between informativeness and redundancy in multimodal representations, the Information Bottleneck (IB) principle has been introduced for multimodal representation learning (Mai et al., 2023a; Wu et al., 2025) to learn features that are strongly correlated with class labels while reducing their correlation with the inputs. Let $X$, $Z$, and $Y$ denote the input features, learned multimodal features, and ground-truth class labels, respectively. The IB principle is enforced by increasing the mutual information between $Z$ and $Y$ while reducing that between $Z$ and $X$, which corresponds to reducing the IB loss $I(Z, X) - I(Z, Y)$, where $I(\cdot, \cdot)$ denotes mutual information. Motivated by this principle, several IB-based multimodal learning methods aim to learn discriminative representations by reducing the IB loss or its variational bound (Mai et al., 2023a; Wu et al., 2025).

**Limitation of Existing Methods.** MIB (Mai et al., 2023a), MCIB (Wang et al., 2026), and OMIB (Wu et al., 2025) explicitly formulate multimodal information bottleneck objectives for fused representations and derive tractable variational upper bounds to explicitly reduce the IB loss. However, they require restrictive and unrealistic assumptions about the latent features of the DNN, which are not applicable to broad DNNs with complex distributions of the latent features. In particular, MIB (Mai et al., 2023a),

[1]School of Computing and Augmented Intelligence, Arizona State University, Tempe, AZ, USA [2]Mayo Clinic Arizona, Arizona, USA. Correspondence to: Yingzhen Yang <yingzhen.yang@asu.edu>.

*Proceedings of the 43rd International Conference on Machine Learning*, Seoul, South Korea. PMLR 306, 2026. Copyright 2026 by the author(s).

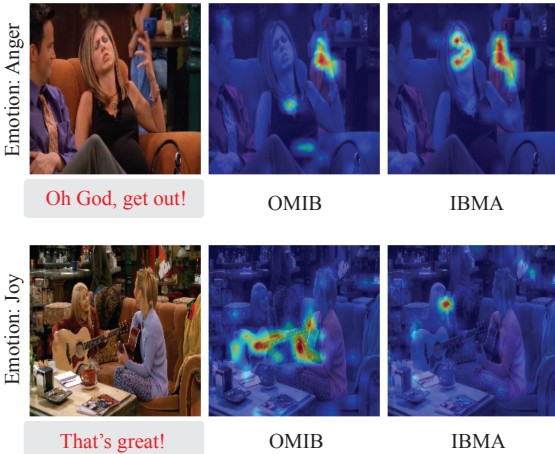

*Figure 1.* Grad-CAM visualization of the image encoder from OMIB and IBMA for two instances from the MELD (Poria et al., 2019) dataset for multimodal emotion recognition (anger in the top and joy in the bottom). The speech scripts from the audio are attached. The speech scripts in both examples clearly convey emotional cues that match the emotions expressed by the speakers. The utterance "Oh God, get out!" reflects strong anger, while "That's great!" clearly expresses joy.

MCIB (Wang et al., 2026), and OMIB (Wu et al., 2025) rely on an unrealistic distributional assumption by adopting a VAE-style surrogate, wherein the conditional distribution $p(\zeta \mid z)$ is enforced to match the marginal $p(\zeta)$ as a fixed Gaussian prior. Herein $z$ denotes the learned task-relevant representation, and $\zeta$ is a latent feature introduced for multimodal representation learning, obtained by encoding $z$ through a VAE encoder. In addition, MIB (Mai et al., 2023a), MCIB (Wang et al., 2026), and OMIB (Wu et al., 2025) apply the IB principle solely to the fused multimodal representation, aiming to learn an informative joint feature.

To address these limitations, we propose Information Bottleneck–based Multimodal Alignment (IBMA), **which makes existing non-applicable IB principles applicable to broad DNNs for multimodal learning**. In a strong contrast to existing methods, the IBMA framework is applicable to arbitrarily complex distributions of the latent features of DNNs due to the novel and distribution-free and computationally efficient upper bound for the IB loss, the IBB, explaining its superior performance over the existing literature, as demonstrated in Tables 2-4 of our paper. IBMA also enforces the IB principle to both the fused multimodal representation and the individual modality-specific representations to learn informative and task-relevant representations with guidance from the other modality. For each modality, IBMA introduces an information-theoretic regularization term that reduces mutual information with the raw input while increasing mutual information with the representation learned from the complementary modality. This design explicitly encourages modality-specific encoders to focus on task-relevant and semantically aligned information, thus enhancing modality-wise feature alignment and leading to improved performance. In addition, we derive a novel, efficient, and distribution-free variational upper bound for the IB loss, termed IBB, which avoids restrictive assumptions on latent feature distributions (Mai et al., 2023a; Wu et al., 2025). The resulting objective is separable across training samples and can be optimized efficiently using standard SGD-based training procedures. We demonstrate the effectiveness of IBMA for multimodal learning tasks, where multimodal data from different domains must be aligned under significant appearance variation and noise.

As illustrated by the Grad-CAM visualization of the image encoders for multimodal emotion recognition in Figure 1, although the speech scripts of both speakers show clear indications of the ground-truth emotion, the state-of-the-art IB-based multimodal learning method, OMIB (Wu et al., 2025), fails to attend to the most emotion-relevant regions in the images. In particular, OMIB either misses critical facial cues, such as the angry face in the top figure and the smiling mouth in the bottom figure, or mistakenly focuses on background regions, such as the guitar and the arm of the other person in the bottom figure, potentially hurting the performance of multimodal emotion recognition. In contrast, our proposed IBMA focuses on the important parts of the human body with clear indications of the speaker's emotion, such as the angry face and gesture of the person in the top figure and the smiling mouth of the person in the bottom figure, guided by the clear indication of the emotion from the speech script of the speaker. This is because the modality-specific representation alignment proposed in IBMA, as explained earlier, enables textual guidance for image representations, which is absent in existing IB-based multimodal learning methods such as OMIB (Wu et al., 2025). Ablation study in Table 4 further demonstrates the advantages of the modality-specific representation alignment. In addition, Table 5 demonstrates the advantages of our efficient and distribution-free variational upper bound for the IB loss over existing upper bounds for the IB loss (Cheng et al., 2020; Dai et al., 2018; Guo et al., 2023). Please refer to Figure 3 in Section E.11 of the appendix for more visualization results on CheXpert (Irvin et al., 2019) for multimodal disease classification.

**Contributions.** The contributions of this paper are presented as follows.

First, we introduce IBMA, a multimodal learning framework that enforces modality-specific representation alignment through the Information Bottleneck principle, explicitly guiding each encoder to learn informative and task-relevant representations with cross-modal supervision. In contrast with existing works, which promote the IB principle solely on the fused representation, IBMA per-

forms modality-specific representation alignment by the modality-specific IB principle to learn informative and task-relevant representations with guidance from the other modality, as well as the reduction of the IB loss on the fused multimodal representation, thus enhancing modality-wise feature alignment with improved performance.

Second, we present a novel and provable variational upper bound for the IB loss, termed IBB, which can be optimized by standard SGD algorithms. The IB loss is then reduced by reducing its upper bound, the IBB. Different from existing upper bounds for the IB loss, such as VIB (Dai et al., 2018; Srivastava et al., 2021), which impose an unrealistic Gaussian assumption on the hidden features of DNNs, and APIB (Guo et al., 2023), which reduces only an approximation to the IB loss, IBMA directly reduces a variational upper bound for the IB loss, IBB, without introducing any distributional assumptions on the hidden features or approximation to the IB loss. Moreover, the proposed IBB is computationally efficient with a computational complexity of $\Theta(nT_0 + nC^2)$, where $C$ is the number of prototypes (to be detailed in Section 3.1), $n$ is the number of training samples, and $T_0$ denotes the computational complexity of a forward and backward pass of the neural network with respect to each training sample. In contrast, the upper bound for the mutual information used to bound the IB loss in CLUB (Cheng et al., 2020), albeit not requiring distributional assumptions on the hidden features of DNNs, requires a substantially higher computational complexity of $\Theta(n^2T_0)$ since $nT_0 \gg C^2$ on multimodal learning datasets detailed in Table 1. A composite loss, which combines the IBB and the regular cross-entropy loss, is used to train DNNs for multimodal learning tasks. Table 4 demonstrates that the IB loss of DNNs trained for multimodal learning can be reduced by optimizing such a composite loss. Table 5 demonstrates that the models using IBB achieve substantially better performance than the models using CLUB, VIB, and APIB. We remark that our variational upper bound, as an independent contribution, can be applied to broader discriminative tasks beyond IB-based multimodal learning. Throughout this paper we use $[N]$ to denote all the natural numbers between 1 and $N$.

## 2. Related Works

**Multimodal Learning.** Early works perform multimodal alignment by leveraging contrastive objectives or attention modules to enforce cross-modal consistency or similarity, thereby projecting heterogeneous modalities into a shared representation space for joint learning (Boecking et al., 2022; Cai et al., 2023; Guo et al., 2024; Mai et al., 2023b; Xie et al., 2025; Zhang et al., 2024a; Zou et al., 2023). However, these methods do not explicitly suppress redundant or noisy information within individual modal-

ities. To explicitly control the trade-off between informativeness and redundancy in multimodal representation, the IB principle has been widely adopted in multimodal learning to learn task-relevant information that is shared across different modalities, while effectively suppressing modality-specific noise and redundant information (Fang et al., 2024; Mai et al., 2023a; Wang et al., 2026; Wu et al., 2025). For instance, DMIB (Fang et al., 2024) proposes a dynamic multimodal fusion framework inspired by the IB principle that preserves label-relevant information via a KL-based sufficiency loss, but it does not explicitly minimize the IB objective, as it increases mutual information with labels without suppressing modality-specific redundancy or noise. MIB (Mai et al., 2023a), MCIB (Wang et al., 2026), and OMIB (Wu et al., 2025) formulate multimodal IB objectives and derive tractable variational upper bounds to directly reduce the IB loss. Nevertheless, these methods rely on a restrictive VAE-style surrogate that enforces the conditional distribution $p(\zeta \mid z)$ to match the marginal $p(\zeta)$, which is further approximated by a fixed Gaussian prior, where $z$ denotes the task-relevant representation and $\zeta$ is a latent variable introduced for multimodal learning.

**Information Bottleneck Principle.** The Information Bottleneck (IB) principle (Tishby et al., 2000) provides a theoretical framework for representation learning by encouraging latent variables to preserve task-relevant information while discarding redundant input details. Deep VIB (Alemi et al., 2017) is the first work to explicitly formulate the IB principle as a training objective for deep neural networks. Motivated by IB, (Lai et al., 2021) introduces a spatial attention mechanism to explicitly reduce the IB loss of attention-weighted features, while (Zhou et al., 2022) shows that self-attention can be interpreted as iterative optimization of the IB objective. Beyond architectural insights, recent theoretical studies (Amjad & Geiger, 2020; Kawaguchi et al., 2023) link IB loss control to improved generalization. Moreover, recent works (Kuang et al., 2023; Wang et al., 2021) also suggest that reducing IB loss enhances adversarial robustness.

## 3. Methods

This section presents the proposed Information Bottleneck-Based Multimodal Alignment (IBMA) framework. We first introduce the motivation and formulation of IBMA by defining modality-specific and multimodal information bottleneck objectives, and derive a novel, distribution-free variational upper bound for the IB loss that is efficient and amenable to SGD optimization in Section 3.1. We then describe the overall IBMA architecture, including modality-specific encoders and a cross-modal attention module, and detail the training procedure of IBMA in Section 3.2.

### 3.1. IBMA: Information Bottleneck-Based Multimodal Alignment

**Motivation.** To learn multimodal representations that are more correlated with the ground-truth label and less correlated with the modality-specific inputs, existing multimodal learning methods based on the IB principle (Mai et al., 2023a; Wu et al., 2025) propose to reduce the multimodal IB loss on the multimodal representation, $\text{IB} = I(Z, X^{(1)}) + I(Z, X^{(2)}) - I(Z, Y)$, where $X^{(j)}$ denotes the random variable representing the input feature in the $j$-th modality for $j \in \{1, 2\}$. In the following text, the domain superscript $j$ takes value in $\{1, 2\}$. $Z$ denotes the random variable representing the multimodal representation, which is obtained by fusing modality-specific representations $Z^{(j)}$ of each domain $j \in \{1, 2\}$. $Y$ denotes the random variable representing the ground-truth training class label. However, existing IB-based methods (Mai et al., 2023a; Wang et al., 2026; Wu et al., 2025) neglect the complementary information encoded in representations learned from different modalities, which could be exploited to learn modality-invariant semantic information while suppressing modality-specific redundancy and noise. To address this issue, we propose a novel multimodal alignment method, Information Bottleneck-Based Multimodal Alignment (IBMA).

IBMA aims to reduce the modality-specific IB loss on the modality-specific representations, $\text{IB}^{(j)} = I(Z^{(j)}, X^{(j)}) - I(Z^{(j)}, Z^{(j')})$, in addition to reducing the multimodal IB loss. In the following text, $j' \neq j$ denotes the index of the modality other than $j$, e.g., $j' = (j \bmod 2) + 1$. Reduction of the modality-specific IB loss ensures that each learned modality-specific representation is more semantically correlated with the modality-specific representation from the other domain and less correlated with the input features. Existing IB-based multimodal learning methods (Mai et al., 2023a; Wu et al., 2025), which do not explicitly reduce the modality-specific IB loss, may retain modality-specific noise or redundant information that is not semantically aligned with the other modality. By reducing the modality-specific IB loss, IBMA explicitly encourages learning task-relevant and modality-invariant representations in each modality, while suppressing modality-specific noise and redundancy. This is achieved by increasing the correlation among learned modality-specific representations across domains while decreasing their correlation with the input features, ultimately leading to more discriminative multimodal representations fused from more discriminative modality-specific representations.

Given the multimodal training data $\{X_i^{(1)}, X_i^{(2)}, Y_i\}_{i=1}^n$ from two different modalities, where $X_i^{(j)}$ stands for the $i$-th input feature from the $j$-th modality, and $Y_i$ is the ground-truth class label for $i \in [n]$, we first specify how to compute the modality-specific IB loss based on the training data. Let $\text{IB}^{(j)} = I(Z^{(j)}, X^{(j)}) - I(Z^{(j)}, Z^{(j')})$ denote the modality-specific IB loss on the $j$-th domain, which treats the representation from the other domain as a cross-modal supervision, guiding $Z^{(j)}$ to retain only the most informative signal of the input from domain $j$ semantically aligned with the other domain $j'$. $X^{(j)}$ takes values in $\{X_i^{(j)}\}_{i=1}^n$. $Z^{(j)}$ takes values in $\{Z_i^{(j)}\}_{i=1}^n$ with $Z_i^{(j)}$ being the $i$-th learned representation by the modality-specific encoder on the $j$-th modality. $Y$ takes values in $\{Y_i\}_{i=1}^n$.

We use prototype-based feature distribution modeling in prototypical learning methods (Caron et al., 2020; Li et al., 2021; Snell et al., 2017; Yang et al., 2018) to model the probability that $Z_i^{(j)}$ belongs to prototype $a$, $\Pr\left[Z^{(j)} \in a\right]$. Suppose we have the learnable prototypes $\left\{\mathcal{F}_a^{(j)}\right\}_{a=1}^C$ for the learned representations $\left\{Z_i^{(j)}\right\}_{i=1}^n$ for the $j$-th domain. Each $\mathcal{F}_a^{(j)}$ is a prototype of the modality-specific representation in prototype $a$, where $C$ is the number of prototypes. Then we define the probability that $Z_i^{(j)}$ belongs to prototype $a$ as $\Pr\left[Z^{(j)} \in a\right] = \frac{1}{n}\sum_{i=1}^n \phi(Z_i^{(j)}, a)$ with

$$\phi(Z_i^{(j)}, a) = \frac{\exp\left(-\left\|Z_i^{(j)} - \mathcal{F}_a^{(j)}\right\|_2^2\right)}{\sum_{a'=1}^C \exp\left(-\left\|Z_i^{(j)} - \mathcal{F}_{a'}^{(j)}\right\|_2^2\right)}.$$

Similarly, we define the joint probabilities

$$\Pr\left[Z^{(j)} \in a, X^{(j)} \in y\right] = \frac{1}{n}\sum_{i=1}^n \phi(Z_i^{(j)} \in a, X_i^{(j)} \in y)$$

and

$$\Pr\left[Z^{(j)} \in a, Z^{(j')} \in y\right] = \frac{1}{n}\sum_{i=1}^n \phi(Z_i^{(j)} \in a, Z_i^{(j')} \in y),$$

where $\phi(Z_i^{(j)} \in a, X_i^{(j)} \in y)$ and $\phi(Z_i^{(j)} \in a, Z_i^{(j')} \in y)$ are computed by (3) and (4) in the appendix using the prototypes.

We note that the computation of the following three probabilities,

$$\Pr\left[Z^{(j)} \in a\right], \Pr\left[Z^{(j)} \in a, X^{(j)} \in y\right],$$
$$\Pr[Z^{(j)} \in a, Z^{(j')} \in y],$$

follows prototype-based feature distribution modeling in prototypical learning methods (Caron et al., 2020; Li et al., 2021; Snell et al., 2017; Yang et al., 2018), which do not require a strong discrete clustering assumption on the

learned features. The success of the prototype-based methods (Snell et al., 2017; Yang et al., 2018) shows that neural networks can produce latent features that are well separated by the prototypes. The information-theoretic objective, IBB, encourages learning latent features aligned with the prototypes. The number of prototypes is larger than the number of classes in each dataset, which is selected by cross-validation (Caron et al., 2020; Li et al., 2021) from $C'$ to $20C'$ with a step of $C'$, where $C'$ is the number of classes. The distribution of the latent feature space is better captured with more prototypes than classes (Caron et al., 2020; Li et al., 2021).

As a result, we can compute the mutual information $I(Z^{(j)}, X^{(j)})$ and $I(Z^{(j)}, Z^{(j')})$ by

$$I(Z^{(j)}, X^{(j)}) = \sum_{a=1}^{C} \sum_{y=1}^{C} \Pr\left[Z^{(j)} \in a, X^{(j)} \in y\right] \log \frac{\Pr\left[Z^{(j)} \in a, X^{(j)} \in y\right]}{\Pr\left[Z^{(j)} \in a\right] \Pr\left[X^{(j)} \in y\right]},$$

$$I(Z^{(j)}, Z^{(j')}) = \sum_{a=1}^{C} \sum_{y=1}^{C} \Pr\left[Z^{(j)} \in a, Z^{(j')} \in y\right] \log \frac{\Pr\left[Z^{(j)} \in a, Z^{(j')} \in y\right]}{\Pr\left[Z^{(j)} \in a\right] \Pr\left[Z^{(j')} \in y\right]},$$

and then compute the IB loss $\text{IB}^{(j)}$. The consistency of the MI estimators is discussed in Section B of the appendix.

Given a variational distribution $Q^{(j)}(Z^{(j)} \in a | Z^{(j')} \in y)$ for $y \in [C]$ and $a \in [C]$, the following theorem gives a variational upper bound, $\text{IBB}^{(j)}$, for the IB loss $\text{IB}^{(j)}$ on the $j$-th domain.

**Theorem 3.1.** Let $\Pr\left[X^{(j)} \in y\right] = \sum_{i=1}^{n} \mathbb{1}_{\{Y_i=y\}}/n := p_y$ for every $y \in [C]$. Then the IB loss on the $j$-th modality satisfies $\text{IB}^{(j)} \leq \text{IBB}^{(j)}$, where the variational upper bound is given by $\text{IBB}^{(j)} := \frac{1}{n} \sum_{i=1}^{n} \left(U_i^{(j)} - V_i^{(j)}\right)$. Here

$$U_i^{(j)} := \sum_{a=1}^{C} \sum_{y=1}^{C} \phi_j(i, a, y) \log\left(\frac{\phi_j(i, a, y)}{p_y \phi(Z_i^{(j)}, a)}\right),$$

$$V_i^{(j)} := \sum_{a=1}^{C} \sum_{y=1}^{C} \phi_{jj'}(i, a, y) \log Q^{(j)}(Z^{(j)} \in a | Z^{(j')} \in y),$$

where $\phi_j(i, a, y) = \phi(Z_i^{(j)} \in a, X_i^{(j)} \in y)$, $\phi_{jj'}(i, a, y) = \phi(Z_i^{(j)} \in a, Z_i^{(j')} \in y)$. $Q^{(j)}(Z^{(j)} \in a | Z^{(j')} \in y)$ is the variational conditional probability that $Z^{(j)}$ belongs to prototype $a$ given $Z^{(j')}$ belongs to class $y$, which is computed efficiently by Algorithm 3 in the supplementary. The proof of Theorem 3.1 follows by applying Lemma C.1 and Lemma C.2 in Section C of the supplementary. We remark that $\text{IBB}^{(j)}$ is ready to be optimized by standard SGD algorithms because it is separable and expressed as the summation of losses on individual training points. Algorithm 1 describes the training process of an IBMA model. The following functions are needed for minibatch-based training with SGD, with the subscript $b$ indicating the corresponding loss on the $b$-th batch $\mathcal{B}_b$:

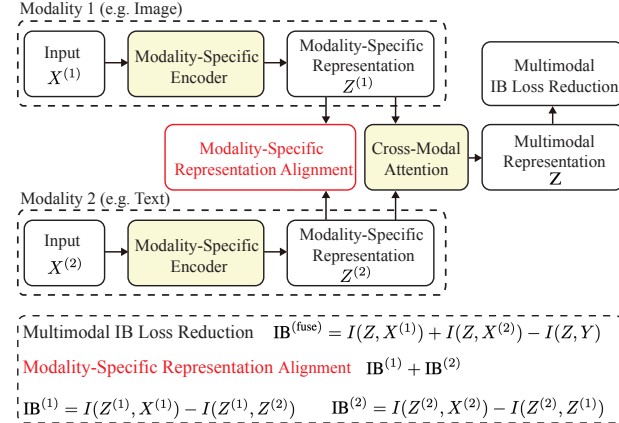

*Figure 2.* IBMA employs modality-specific encoders to learn representations from each input modality, such as image and text. A cross-modal attention module integrates the modality-specific representations to form a fused multimodal representation. In addition to reducing the multimodal IB loss on the fused multimodal representation, IBMA further enforces modality-specific representation alignment by leveraging representations from other modalities as cross-modal supervision, which ensures the learning of semantically aligned modality-specific representations.

$\text{IBB}_b^{(j)} = \frac{1}{|\mathcal{B}_b|} \sum_{i \in \mathcal{B}_b} \left(U_i^{(j)} - V_i^{(j)}\right)$. In addition to reducing the modality-specific IB loss $\text{IB}^{(j)}$ by reducing $\text{IBB}^{(j)}$, IBMA also reduces the multimodal IB loss on the multimodal representation $Z$ fused from the modality-specific representations. Let $\text{IB}^{(\text{fuse})} = I(Z, X^{(1)}) + I(Z, X^{(2)}) - I(Z, Y)$ be the multimodal IB loss on the multimodal representation $Z$, where $Y$ denotes the class label taking value in $\{Y_i\}_{i=1}^{n}$. $\text{IBB}^{(\text{fuse})}$ denotes the variational upper bound for $\text{IB}^{(\text{fuse})}$ which is derived in a similar manner as in Theorem 3.1 and to be detailed in Section C.2 of the appendix. Here $Z$ takes values in $\{Z_i\}_{i=1}^{n}$ and $Z_i$ denotes the fused multimodal representation for the $i$-th input. We compute prototypes $\{\mathcal{F}_a\}_{a=1}^{C}$ on the fused multimodal representation, where each $\mathcal{F}_a$ is the average of the fused multimodal representation in prototype $a$. We also define the probability that $Z_i$ belongs to prototype $a$ as $\Pr[Z \in a] = \frac{1}{n} \sum_{i=1}^{n} \phi(Z_i, a)$ similar to the computation of $\Pr\left[Z^{(j)} \in a\right]$. The training loss of IBMA on the $b$-th batch is then computed by

$$\mathcal{L}_b = \text{CE}_b + \eta \text{IBB}_b, \tag{1}$$

where $\text{IBB}_b = \text{IBB}_b^{(1)} + \text{IBB}_b^{(2)} + \text{IBB}_b^{(\text{fuse})}$ and $\text{CE}_b = \text{CE}_b^{(1)} + \text{CE}_b^{(2)} + \text{CE}_b^{(\text{fuse})}$. $\text{CE}_b^{(j)} = \frac{1}{|\mathcal{B}_b|} \sum_{i \in \mathcal{B}_b} H(Z_i^{(j)}, Y_i)$ is the cross-entropy loss on the $b$-th batch $\mathcal{B}_b$ in the $j$-th domain. $\text{CE}_b^{(\text{fuse})} = \frac{1}{|\mathcal{B}_b|} \sum_{i \in \mathcal{B}_b} H(Z_i, Y_i)$ is the cross-entropy loss on the $b$-th batch $\mathcal{B}_b$ for the fused multimodal representation. $H(\cdot, \cdot)$ is the cross-entropy function. $\eta$ is the balance factor for IBB. The values of $\eta$ on different datasets will be decided by performing cross-validation.

**Efficient and Distribution-Free Variational Upper Bound.** Because our variational upper bounds are distribution-free, we demonstrate the computational efficiency of $\text{IBB}^{(j)}$ over the existing distribution-free upper bound for the IB loss in CLUB (Cheng et al., 2020), and a similar conclusion holds for $\text{IBB}^{(\text{fuse})}$. Let $T_0$ denote the computational complexity of a forward and backward pass of the neural network with respect to each training sample. The overall computational complexity for calculating the proposed modality-specific IBB regularization term $\text{IBB}^{(j)}$ is $\Theta(nT_0 + nC^2)$. In contrast, computing the upper bound for the mutual information required for calculating the upper bound for the IB loss proposed in CLUB (Cheng et al., 2020) requires a substantially higher computational complexity at least $\Theta(n^2 T_0)$, since $nT_0 \gg C^2$ on multimodal learning datasets detailed in Table 1. We note that $\Theta(n^2 T_0)$ corresponds exclusively to the upper bound for the mutual information $I(Z, X^{(j)})$, while CLUB additionally requires the computation of the lower bound for the mutual information $I(Z, Y)$. Details on the complexity analysis of CLUB and our IBB are presented in Section F of the supplementary. Importantly, in contrast to existing IB upper bounds that rely on the unrealistic Gaussian distributional assumption of hidden features (Dai et al., 2018; Srivastava et al., 2021) or on the approximation to the IB loss (Guo et al., 2023), our bound for the IB loss, the IBB, does not require any distributional assumptions on the hidden features or approximation to the IB loss. Table 5 in Section 4.5 demonstrates that IBB achieves substantially better performance than CLUB and other competing methods that rely on the unrealistic Gaussian distribution assumption of hidden features (Dai et al., 2018; Srivastava et al., 2021) or on the approximation to the IB loss (Guo et al., 2023), and requires much less training time than the distribution-free baseline method, CLUB (Cheng et al., 2020).

### 3.2. IBMA Architecture and Training Procedure

The proposed IBMA framework comprises two main components, namely modality-specific encoders that generate modality-specific representations and a cross-modal attention module that fuses modality-specific representations into a multimodal representation. An overview of the architecture is illustrated in Figure 2.

**Model Architecture of IBMA.** Given the input $X_i^{(j)}$ from the $j$-th domain for the $i$-th input, IBMA employs a modality-specific encoder $f^{(j)}(\cdot)$ to extract modality-specific representation $Z_i^{(j)} = f^{(j)}(X_i^{(j)})$. Each modality-specific encoder is implemented using a modality-specific backbone, such as ResNet (He et al., 2016) for image data and BERT (Devlin et al., 2019) for text data. The modality-specific representations $Z_i^{(1)}$ and $Z_i^{(2)}$ for the $i$-th input are fused using a cross-modal attention module widely used

in the multimodal learning literature (Wang et al., 2026; Wu et al., 2025), which produces a joint multimodal representation $Z_i = F(Z_i^{(1)}, Z_i^{(2)})$, where $F(\cdot, \cdot)$ denotes the cross-modal attention module. The fused representation $Z_i$ for the $i$-th input is used for the downstream tasks. IBMA naturally extends to tasks with more than two modalities by applying a modality-specific encoder to each modality and performing cross-modal attention–based fusion jointly across multiple modalities following (Mai et al., 2023a; Wu et al., 2025), while applying modality-specific representation alignment to each pair of modalities.

**Training Procedure of IBMA.** IBMA is trained end-to-end by jointly optimizing the modality-specific encoders and the cross-modal fusion module. The network parameters are updated by optimizing the joint training loss $\mathcal{L}_b$ in Equation (1) for the training data in the $b$-th batch. Algorithm 1 in Section A of the appendix describes the training of the IBMA model. Such a joint optimization strategy ensures that IBMA simultaneously achieves modality-wise feature alignment and effective multimodal fusion under the guidance of the IB principle.

## 4. Experiment

In this section, we evaluate the proposed IBMA across multiple multimodal learning tasks. Section 4.1 first describes the datasets and experimental settings throughout the experiments. Section 4.2 evaluates IBMA for multimodal emotion recognition. Section 4.3 evaluates the performance of IBMA on multimodal disease classification on two large-scale datasets, MIMIC-CXR (Johnson et al., 2019) and CheXpert (Irvin et al., 2019). Section 4.4 presents an ablation study analyzing the impact of modality-specific representation alignment and the proposed IBB in IBMA. Section 4.5 compares the proposed IBB variational upper bound for the IB loss with existing IB upper-bound formulations in terms of both effectiveness and training efficiency. Additional experiment results are presented in Section E of the appendix. Section E.1 evaluates the performance of IBMA for multimodal sentiment analysis. Section E.2 evaluates the effectiveness of IBMA for multimodal anomalous tissue detection. We evaluate IBMA for multimodal classification on PME4 (Chen et al., 2022) with more than three modalities in Section E.3. We compare the modality-specific IB alignment in IBMA with contrastive-based and distillation-based multimodal alignment methods in Section E.4. We demonstrate the effectiveness and the training efficiency of IBMA on multimodal datasets with a larger number of classes, including UPMC Food-101 (Wang et al., 2015) and WIKI-DOC (Fujinuma et al., 2023), in Section E.5. We analyze the sensitivity of IBMA to the choice of the balancing factor $\eta$ in the joint training loss and the prototype number $C$ in Section E.6.

We study the effectiveness of IBMA for self-supervised multimodal pre-training in Section E.7. Section E.8 reports the training time comparisons between IBMA and the competing multimodal learning baselines. Section E.9 evaluates the statistical significance of the observed performance improvements of IBMA. We study the classification performance using the modality-specific encoder alone in IBMA in Section E.10. The Grad-CAM visualization analysis on CheXpert for multimodal disease classification is presented in Section E.11. We perform t-sne visualization analysis on representations learned by IBMA in Section E.12.

### 4.1. Implementation Details

We evaluate IBMA on various multimodal benchmarks, including emotion recognition datasets, CREMA-D (Cao et al., 2014), MELD (Poria et al., 2019), and IEMO-CAP (Busso et al., 2008), sentiment analysis datasets, CMU-MOSI (Zadeh et al., 2016) and MELD (Poria et al., 2019), pathological tissue datasets, 10x-hNB-A–H and 10x-hBC-A–D (Xu et al., 2024), and disease classification datasets, MIMIC-CXR (Johnson et al., 2019) and CheXpert (Irvin et al., 2019). Five common thorax diseases from MIMIC-CXR and CheXpert are used following the literature on thorax disease classification (Xiao et al., 2023). UPMC Food-101 (Wang et al., 2015) and WIKI-DOC (Fujinuma et al., 2023) are datasets with a large number of classes. PME4 is a four-modality classification dataset (Chen et al., 2022). Detailed dataset descriptions are provided in Section D.1 of the appendix, and Table 1 summarizes the training and test splits for all datasets. It is worthwhile to mention that MELD, MIMIC-CXR, and CheXpert are highly imbalanced datasets. For example, the neutral class with 4710 training samples is $17.4\times$ and $17.6\times$ larger than disgust with 271 training samples and fear with 268 training samples in MELD. Detailed experimental settings and descriptions of the compared methods are provided in Section D.2 and D.3 of the appendix.

Table 1. Summary of the datasets.

| Dataset | # Classes | Training size | Test size |
|---|---|---|---|
| CREMA-D (Cao et al., 2014) | 6 | 6,698 | 744 |
| CMU-MOSI (Zadeh et al., 2016) | 7 | 1,281 | 685 |
| 10x-hNB & 10x-hBC (Xu et al., 2024) | 2 | 18,183 | 1,123 |
| IEMOCAP (Busso et al., 2008) | 5 | 5,990 | 1,541 |
| MELD (Poria et al., 2019) | 7 | 9,989 | 2,610 |
| MIMIC-CXR (Johnson et al., 2019) | 5 | 227,835 | 1,000 |
| CheXpert (Irvin et al., 2019) | 5 | 223,414 | 234 |
| PME4 (Chen et al., 2022) | 7 | 4,800 | 1,200 |
| UPMC Food-101 (Wang et al., 2015) | 101 | 67,988 | 22,716 |
| WIKI-DOC (Fujinuma et al., 2023) | 111 | 54,808 | 4,988 |

### 4.2. Multimodal Emotion Recognition

We evaluate IBMA on multimodal emotion recognition, where emotional states are jointly predicted from heterogeneous modalities, following standard protocols in the multimodal learning literature (Cui et al., 2024; Mai et al., 2023a; Wu et al., 2025). For CREMA-D, we adopt

the audio–visual setting in (Wu et al., 2025), where raw speech waveforms are converted into log-Mel spectrograms and encoded using a ResNet-18 backbone to obtain fixed-dimensional acoustic embeddings, while facial video frames are processed by a shared-weight ResNet-18 encoder and temporally averaged to produce utterance-level visual representations. For MELD and IEMOCAP, we adopt a tri-modal setting with audio, visual, and textual modalities following (Mai et al., 2023a), where modality-specific representation alignment is applied to each pair of modalities. The textual modality is encoded using a BERT-based text encoder (Liu et al., 2019). The resulting modality-specific representations are fused via a cross-modal attention module to produce a unified multimodal representation for emotion prediction. It is observed in Table 2 that IBMA consistently outperforms existing multimodal learning methods across all three benchmarks. In particular, IBMA outperforms the strongest baseline by 1.8% on CREMA-D, demonstrating the effectiveness of modality-specific representation alignment.

Table 2. Accuracy comparison for emotion recognition on CREMA-D, MELD, and IEMOCAP. The IBMA results are averaged over 10 runs with different random initializations and reported as mean ± standard deviation, while the statistical significance of its improvements over the best baseline is presented in Table 17 in Section E.9 of the appendix.

| Method | CREMA-D | MELD | IEMOCAP |
|---|---|---|---|
| Concat (Perez et al., 2018) | 53.2 | 56.1 | 67.4 |
| BiGated (Kiela et al., 2018) | 58.4 | 59.8 | 70.2 |
| MISA (Hazarika et al., 2020) | 57.7 | 60.5 | 71.0 |
| Deep IB (Wang et al., 2019) | 54.1 | 57.3 | 68.6 |
| MMIB (Zhang et al., 2022) | 56.7 | 61.4 | 72.3 |
| MMRLIB (Cui et al., 2024) | 57.3 | 62.0 | 73.1 |
| MIB (Mai et al., 2023a) | 61.4 | 63.6 | 74.1 |
| MCIB (Wang et al., 2026) | 63.2 | 64.0 | 74.3 |
| CLFA (Zhang et al., 2024a) | 63.0 | 63.5 | 74.0 |
| DCLF (Xie et al., 2025) | 63.4 | 63.2 | 74.2 |
| OMIB (Wu et al., 2025) | 63.6 | 64.3 | 74.2 |
| **IBMA (Ours)** | **65.4**±0.4 | **66.3**±0.3 | **75.7**±0.3 |

Table 3. mAUC comparison for disease classification on MIMIC-CXR and CheXpert. The IBMA results are averaged over 10 runs with different random initializations and reported as mean ± standard deviation, while the statistical significance of its improvements over the best baseline is presented in Table 19 in Section E.9 of the appendix.

| Method | MIMIC-CXR | CheXpert |
|---|---|---|
| Concat (Perez et al., 2018) | 67.9 | 85.0 |
| BiGated (Kiela et al., 2018) | 68.4 | 87.5 |
| MISA (Hazarika et al., 2020) | 68.2 | 86.4 |
| Deep IB (Wang et al., 2019) | 67.0 | 86.2 |
| MMIB (Zhang et al., 2022) | 69.6 | 87.7 |
| MMRLIB (Cui et al., 2024) | 69.4 | 87.5 |
| MIB (Mai et al., 2023a) | 70.2 | 88.1 |
| MCIB (Wang et al., 2026) | 70.6 | 88.4 |
| CLFA (Zhang et al., 2024a) | 70.5 | 88.6 |
| DCLF (Xie et al., 2025) | 71.0 | 88.9 |
| OMIB (Wu et al., 2025) | 71.0 | 89.3 |
| **IBMA (Ours)** | **72.7**±0.3 | **91.1**±0.2 |

*Table 4.* Ablation study on the contributions of modality-specific representation alignment and IBB for multimodal disease classification. Lower (more negative) values of IB loss indicate stronger IB alignment and better adherence to the IB principle. The IBMA results are averaged over 10 runs with different random initializations and reported as mean $\pm$ standard deviation.

| Dataset | Method | Visual IB Loss ↓ | Textual IB Loss ↓ | Multimodal IB Loss ↓ | mAUC (%) ↑ |
|---|---|---|---|---|---|
| MIMIC-CXR | OMIB (Wu et al., 2025) | -0.021 | -0.024 | -0.028 | 71.0 |
| | IBMA w/o Modality-Specific Representation Alignment | -0.028 | -0.031 | -0.035 | 71.4 |
| | IBMA w/o IBB | -0.036 | -0.036 | -0.044 | 71.7 |
| | **IBMA (Ours)** | **-0.048** | **-0.045** | **-0.058** | **72.7**$\pm$0.3 |
| CheXpert | OMIB (Wu et al., 2025) | -0.020 | -0.023 | -0.028 | 89.3 |
| | IBMA w/o Modality-Specific Representation Alignment | -0.027 | -0.030 | -0.036 | 89.7 |
| | IBMA w/o IBB | -0.035 | -0.035 | -0.043 | 90.0 |
| | **IBMA (Ours)** | **-0.049** | **-0.043** | **-0.055** | **91.1**$\pm$0.2 |

## 4.3. Multimodal Disease Classification

We evaluate IBMA for multimodal thoracic disease classification on the large-scale medical vision-language benchmarks MIMIC-CXR (Johnson et al., 2019) and CheXpert (Irvin et al., 2019), where the goal is to predict the presence of multiple thoracic pathologies from the chest X-ray and the clinical reports. Chest X-ray images are encoded with a ViT-B (Xiao et al., 2023) encoder and radiology reports with a BERT-based text encoder (Huang et al., 2019) pre-trained for medical predictions. For both MIMIC-CXR and CheXpert, we report mean AUROC (mAUC), defined as the arithmetic mean of per-disease AUROC scores across all diseases. It is observed in Table 3 that IBMA significantly outperforms all competing multimodal learning methods. For instance, IBMA outperforms OMIB by 1.4% in mAUC on CheXpert, demonstrating the advantages of IBMA.

## 4.4. Ablation Study on the Modality-Specific Representation Alignment

To study the effect of modality-specific representation alignment in IBMA, we construct an ablation model, termed IBMA w/o Modality-Specific Representation Alignment, which removes all modality-specific IB loss while retaining the same modality-specific encoders, cross-modal fusion module, and multimodal IB loss on the fused multimodal representation. To further compare the contribution of the proposed IBB with that of modality-specific IB alignment, we also consider an additional ablation model, termed IBMA w/o IBB, which replaces the proposed IBB with the existing upper bound CLUB (Cheng et al., 2020) while keeping the same modality-specific encoders and fusion architecture. We compare IBMA with the ablation models and OMIB (Wu et al., 2025) in terms of modality-specific IB loss, multimodal IB loss, and the mAUC for multimodal disease classification. As shown in Table 4, both the modality-specific IB alignment and the proposed IBB contribute to the superior performance of IBMA. In particular, IBMA renders significantly lower modality-specific IB loss than OMIB and the ablation model without modality-specific representation alignment,

leading to a lower multimodal IB loss and better classification accuracy. Moreover, replacing IBB with CLUB also degrades the performance on both datasets. For example, on CheXpert, removing the modality-specific IB alignment or replacing IBB with CLUB leads to 1.0% and 0.7% drops in mAUC, respectively, compared with the full IBMA model.

## 4.5. Comparison between IBB and Existing Upper Bounds for the IB Loss

We compare the proposed variational upper bound for the IB loss, IBB, with existing works deriving the upper bound for the IB loss (Cheng et al., 2020; Dai et al., 2018; Srivastava et al., 2021), including CLUB (Cheng et al., 2020) and VIB (Dai et al., 2018; Srivastava et al., 2021). We also compare IBB with APIB (Guo et al., 2023), which relies on IB approximation instead of directly reducing the IB loss. The comparison is conducted by replacing IBB

*Table 5.* Comparison of different methods for reducing the IB loss on CREMA-D for emotion recognition. The IBB results are averaged over 10 runs with different random initializations and reported as mean $\pm$ standard deviation, while the statistical significance of its improvements over the best baseline is presented in Table 21 in Section E.9 of the appendix.

| Methods | Training Time (Min/Epoch) | Accuracy |
|---|---|---|
| VIB (Dai et al., 2018) | 1.9 | 63.9 |
| APIB (Guo et al., 2023) | 1.9 | 63.8 |
| CLUB (Cheng et al., 2020) | 3.8 | 64.1 |
| **IBB (Ours)** | 2.0 | **65.4**$\pm$0.4 |

with VIB, APIB, and CLUB in IBMA for emotion recognition on CREMA-D, with training time reported on a single NVIDIA A100 GPU. It is observed in Table 5 that the unrealistic Gaussian distribution assumption on the hidden features imposed by VIB (Dai et al., 2018) and the IB approximation in APIB (Guo et al., 2023) lead to degraded performance compared with IBB. For example, the IBB-based model outperforms the VIB-based model by 1.5% in accuracy on CREMA-D, with only a marginal 5.3% increase in training time. Moreover, the IBB-based model outperforms the CLUB-based model by 1.3% in accuracy while using only 52.6% of its training time.

# 5. Conclusion

In this paper, we propose Information Bottleneck–based Multimodal Alignment (IBMA), a novel multimodal learning framework that enforces the IB principle at both the fused multimodal and modality-specific representation levels. By introducing modality-specific representation alignment, IBMA explicitly guides each modality-specific encoder to learn task-relevant and semantically aligned representations with cross-modal supervision, while effectively suppressing modality-dependent noise and redundancy. In addition, we derive a novel, efficient, and distribution-free variational upper bound for the IB loss, IBB. Extensive experiments and ablation studies across diverse multimodal benchmarks demonstrate that IBMA significantly outperforms existing multimodal learning methods.

# Acknowledgements

This work was supported by National Institute of Health 1OT2OD037955-01. This work was also supported by the 2023 Mayo Clinic and Arizona State University Alliance for Health Care Collaborative Research Seed Grant Program under Award No. AWD00038846.

# Impact Statement

This paper aims to advance multimodal representation learning by improving cross-modal alignment and reducing modality-specific noise and redundancy. IBMA may benefit applications such as medical image–text analysis, multimodal emotion recognition, and other decision-support systems where robust integration of heterogeneous data is important. However, deployment in sensitive domains requires caution, as models may inherit dataset biases, perform unevenly across demographic groups or institutions, and raise privacy or consent concerns when using clinical, facial, speech, or textual data. IBMA should therefore be used with appropriate human oversight, subgroup evaluation, privacy-preserving data handling, and careful validation under distribution shift, and is not intended for surveillance, unauthorized profiling, or fully automated high-stakes decision-making.

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

## A. Training Algorithm of IBMA

Algorithm 1 describes the training of the IBMA model.

---

**Algorithm 1** Training Algorithm of the IBMA Model

---

**Require:** Multimodal training data $\left\{ X_i^{(1)}, X_i^{(2)}, Y_i \right\}_{i=1}^n$, mini-batches $\{\mathcal{B}_b\}_{b=1}^B$, total training epochs $t_{\text{train}}$, learning rate $\alpha$.
**Ensure:** Trained network parameters $\mathcal{W}$.
1: Initialize the network parameters $\mathcal{W}$ randomly.
2: **for** $j = 1$ to $2$ **do**
3:     Compute initial prototypes $\left\{ \mathcal{C}_y^{(j)} \right\}_{y=1}^C$.
4:     $j' = (j \bmod 2) + 1$
5:     Initialize the variational distribution $Q^{(j)}(Z^{(j)} \in a \mid Z^{(j')} \in y) = \frac{1}{C}$ for all $a, y \in [C]$.
6: **end for**
7: Initialize the fused variational distribution $Q(Z \in a \mid Y = y) = \frac{1}{C}$ for all $a, y \in [C]$.
8: Initialize fused prototypes $\{\mathcal{F}_a\}_{a=1}^C$ for $Z$ as zeros or using one forward pass.
9: **for** $t = 1$ to $t_{\text{train}}$ **do**
10:     **for** $b = 1$ to $B$ **do**
11:       Update $\mathcal{W} \leftarrow \mathcal{W} - \alpha \nabla_{\mathcal{W}} \mathcal{L}_b$ using mini-batch $\mathcal{B}_b$, where $\mathcal{L}_b$ is defined in Equation (1).
12:     **end for**
13:     **for** $j = 1$ to $2$ **do**
14:       $j' = (j \bmod 2) + 1$
15:       Update $Q^{(j)}(Z^{(j)} \in a \mid Z^{(j')} \in c)$ using Algorithm 3.
16:       Update prototypes $\left\{ \mathcal{F}_a^{(j)} \right\}_{a=1}^C$.
17:     **end for**
18:     Update $Q(Z \in a \mid Y = y)$ using Algorithm 4.
19:     Update prototypes $\{\mathcal{F}_a\}_{a=1}^C$.
20: **end for**
21: **return** Trained network parameters $\mathcal{W}$.

---

## B. Consistency of the Mutual Information Estimators

We now discuss the accuracy (consistency) of the MI estimators in Section 3.1. For a fixed modality $j \in \{1, 2\}$, let $j' \neq j$ denote the other modality. The consistency of such estimators depends on the consistency of the estimator $\Pr\left[ Z^{(j)} \in a \right]$ for the probability that the learned representation $Z^{(j)}$ belongs to prototype $a$, and the consistency of the estimators $\Pr\left[ Z^{(j)} \in a, X^{(j)} \in y \right]$ and $\Pr\left[ Z^{(j)} \in a, Z^{(j')} \in y \right]$ for the corresponding ground-truth joint probabilities. We now analyze the consistency of these probability estimators.

The estimator $\Pr\left[ Z^{(j)} \in a \right] = \frac{1}{n} \sum_{i=1}^n \phi(Z_i^{(j)}, a)$ is a consistent plugin estimator. Let $\mathcal{F}^{(j)} = \left\{ \mathcal{F}_k^{(j)} \right\}_{k=1}^C$ denotes the set of $C$ learnable prototypes for the learned representations in the $j$-th modality. According to the theory of asymptotic quantization in (Pollard, 1982), for a fixed $C$, the prototypes $\mathcal{F}^{(j)}$ optimized via empirical risk minimization converge almost surely to the optimal prototypes $\mathcal{F}^{(j)\star}$ that minimize the expected quantization error:

$$\lim_{n \to \infty} \min_{\mathcal{F}^{(j)}} \frac{1}{n} \sum_{i=1}^n \left\| Z_i^{(j)} - \mathcal{F}_k^{(j)} \right\|_2^2 = \min_{\mathcal{F}^{(j)}} \mathbb{E}\left[ \left\| Z^{(j)} - \mathcal{F}_k^{(j)} \right\|_2^2 \right], \quad j \in \{1, 2\}. \tag{2}$$

We note that each $\mathcal{F}_k^{(j)}$ is computed as the weighted average of the learned features in prototype $k$ for all $k \in [C]$, so $\mathcal{F}^{(j)}$ minimizes the left-hand side of (2), guaranteeing the consistency of $\mathcal{F}^{(j)}$.

We note that the asymptotic result (2) can be converted to a finite-sample result presented as follows. We define

$$R_n(\mathcal{F}^{(j)}) := \frac{1}{n} \sum_{i=1}^n \| Z_i^{(j)} - c \|^2, \qquad R(\mathcal{F}^{(j)}) := \mathbb{E}\left[ \left\| Z^{(j)} - c \right\|_2^2 \right],$$

and let

$$\widehat{\mathcal{F}}_n^{(j)} \in \arg \min_{|\mathcal{F}^{(j)}|=C} R_n(\mathcal{F}^{(j)}), \qquad \mathcal{F}^{(j)\star} \in \arg \min_{|\mathcal{F}^{(j)}|=C} R(\mathcal{F}^{(j)}).$$

Then, under standard regularity conditions, for any $\delta \in (0, 1)$, with probability at least $1 - \delta$, we have

$$R(\widehat{\mathcal{F}}_n^{(j)}) - R(\mathcal{F}^{(j)\star}) \leq \varepsilon_{n,\delta},$$

where $\varepsilon_{n,\delta} = O\left(\sqrt{\frac{1}{n} \log \frac{1}{\delta}}\right)$.

Furthermore, under the softmax formulation, $\phi(Z_i^{(j)}, a)$ acts as a continuous relaxation of the hard assignment. The consistency of the estimator $\Pr\left[Z^{(j)} \in a\right]$ then follows from the standard kernel density estimation literature (EINMAHL & MASON, 2005; Goldstein & Messer, 1992; RIGOLLET & VERT, 2009), together with the consistency result in Eq. (2) above.

Similarly, with the estimator

$$\Pr\left[Z^{(j)} \in a, X^{(j)} \in y\right] = \frac{1}{n}\sum_{i=1}^n \phi(Z_i^{(j)} \in a, X_i^{(j)} \in y),$$

where $\phi(Z_i^{(j)} \in a, X_i^{(j)} \in y)$ is defined in (3), it can be shown that $\Pr\left[Z^{(j)} \in a, X^{(j)} \in y\right]$ converges to the ground-truth joint probability that $Z^{(j)} \in a, X^{(j)} \in y$.

Likewise, with the estimator

$$\Pr\left[Z^{(j)} \in a, Z^{(j')} \in y\right] = \frac{1}{n}\sum_{i=1}^n \phi(Z_i^{(j)} \in a, Z_i^{(j')} \in y),$$

where

$$\phi(Z_i^{(j)} \in a, Z_i^{(j')} \in y) = \frac{\exp\left(-\left\|Z_i^{(j)} - \mathcal{F}_a^{(j)}\right\|_2^2 - \left\|Z_i^{(j')} - \mathcal{F}_y^{(j')}\right\|_2^2\right)}{\sum_{a'=1}^C \sum_{y'=1}^C \exp\left(-\left\|Z_i^{(j)} - \mathcal{F}_{a'}^{(j)}\right\|_2^2 - \left\|Z_i^{(j')} - \mathcal{F}_{y'}^{(j')}\right\|_2^2\right)},$$

which is also specified in (4) of this appendix, it can be shown that $\Pr\left[Z^{(j)} \in a, Z^{(j')} \in y\right]$ converges to the ground-truth joint probability that $Z^{(j)} \in a, Z^{(j')} \in y$.

The consistency of the three estimators, $\Pr\left[Z^{(j)} \in a\right]$, $\Pr\left[Z^{(j)} \in a, X^{(j)} \in y\right]$, and $\Pr\left[Z^{(j)} \in a, Z^{(j')} \in y\right]$, guarantees the consistency of the estimators for the two MI terms, $I(Z^{(j)}, X^{(j)})$ and $I(Z^{(j)}, Z^{(j')})$.

## C. Proof of Theorem 3.1

We define

$$\phi\left(Z_i^{(j)} \in a, X_i^{(j)} \in y\right) = \frac{\exp\left(-\left\|Z_i^{(j)} - \mathcal{F}_a^{(j)}\right\|_2^2 - \left\|X_i^{(j)} - \mathcal{C}_y^{(j)}\right\|_2^2\right)}{\sum_{a'=1}^C \sum_{y'=1}^C \exp\left(-\left\|Z_i^{(j)} - \mathcal{F}_{a'}^{(j)}\right\|_2^2 - \left\|X_i^{(j)} - \mathcal{C}_{y'}^{(j)}\right\|_2^2\right)}, a \in [C], y \in [C], \tag{3}$$

$$\phi\left(Z_i^{(j)} \in a, Z_i^{(j')} \in y\right) = \frac{\exp\left(-\left\|Z_i^{(j)} - \mathcal{F}_a^{(j)}\right\|_2^2 - \left\|Z_i^{(j')} - \mathcal{F}_y^{(j')}\right\|_2^2\right)}{\sum_{a'=1}^C \sum_{y'=1}^C \exp\left(-\left\|Z_i^{(j)} - \mathcal{F}_{a'}^{(j)}\right\|_2^2 - \left\|Z_i^{(j')} - \mathcal{F}_{y'}^{(j')}\right\|_2^2\right)}, a \in [C], y \in [C]. \tag{4}$$

Here $\left\{\mathcal{C}_y^{(j)}\right\}_{y=1}^C$ are the learnable prototypes of the input features $\left\{X_i^{(j)}\right\}_{i=1}^n$ for the $j$-th domain.

**Lemma C.1.** Let $\Pr\left[X^{(j)} \in y\right] = \sum_{i=1}^n \mathbb{I}_{\{Y_i = y\}}/n := p_y$ for every $y \in [C]$, then

$$I(Z^{(j)}, X^{(j)}) \leq \frac{1}{n}\sum_{i=1}^n \sum_{a=1}^C \sum_{y=1}^C \phi\left(Z_i^{(j)} \in a, X_i^{(j)} \in y\right) \log\left(\frac{\phi\left(Z_i^{(j)} \in a, X_i^{(j)} \in y\right)}{p_y \phi(Z_i^{(j)}, a)}\right). \tag{5}$$

*Proof.* Then the joint probability $\Pr\left[Z^{(j)} \in a, X^{(j)} \in y\right]$ is specified by

$$\Pr\left[Z^{(j)} \in a, X^{(j)} \in y\right] = \frac{1}{n}\sum_{i=1}^{n} \phi\left(Z_i^{(j)} \in a, X_i^{(j)} \in y\right). \tag{6}$$

It then follows from (6) and the log sum inequality that

$$
\begin{aligned}
I(Z^{(j)}, X^{(j)}) &= \sum_{a=1}^{C}\sum_{y=1}^{C}\Pr\left[Z^{(j)} \in a, X^{(j)} \in y\right]\log\frac{\Pr\left[Z^{(j)} \in a, X^{(j)} \in y\right]}{\Pr\left[Z^{(j)} \in a\right]\Pr\left[X^{(j)} \in y\right]} \\
&\leq \frac{1}{n}\sum_{i=1}^{n}\sum_{a=1}^{C}\sum_{y=1}^{C}\phi\left(Z_i^{(j)} \in a, X_i^{(j)} \in y\right)\log\left(\phi\left(Z^{(j)} \in a, X^{(j)} \in y\right)\right) \\
&\quad -\frac{1}{n}\sum_{i=1}^{n}\sum_{a=1}^{C}\sum_{y=1}^{C}\phi\left(Z_i^{(j)} \in a, X_i^{(j)} \in y\right)\log\left(\phi(Z_i^{(j)}, a)p_y\right) \\
&= \frac{1}{n}\sum_{i=1}^{n}\sum_{a=1}^{C}\sum_{y=1}^{C}\phi\left(Z_i^{(j)} \in a, X_i^{(j)} \in y\right)\log\left(\frac{\phi\left(Z_i^{(j)} \in a, X_i^{(j)} \in y\right)}{p_y\phi(Z_i^{(j)}, a)}\right). \tag{7}
\end{aligned}
$$

$\square$

**Lemma C.2.** Recall that $j' = (j \bmod 2) + 1$ is the modality other than $j$, we have

$$I(Z^{(j)}, Z^{(j')}) \geq \frac{1}{n}\sum_{i=1}^{n}\sum_{a=1}^{C}\sum_{y=1}^{C}\phi\left(Z_i^{(j)} \in a, Z_i^{(j')} \in y\right)\log Q^{(j)}(Z^{(j)} \in a | Z^{(j')} \in y). \tag{8}$$

*Proof.* Let $Q^{(j)}(Z^{(j)}|Z^{(j')})$ be a variational distribution. We have

$$
\begin{aligned}
I(Z^{(j)}, Z^{(j')}) &= \sum_{a=1}^{C}\sum_{y=1}^{C}\Pr\left[Z^{(j)} \in a, Z^{(j')} \in y\right]\log\frac{\Pr\left[Z^{(j)} \in a, Z^{(j')} \in y\right]}{\Pr[Z^{(j)} \in a]\Pr[Z^{(j')} \in y]} \\
&= \sum_{a=1}^{C}\sum_{y=1}^{C}\Pr\left[Z^{(j)} \in a, Z^{(j')} \in y\right]\log\frac{\Pr\left[Z^{(j)} \in a|Z^{(j')} \in y\right]Q^{(j)}(Z^{(j)} \in a|Z^{(j')} \in y)}{\Pr[Z^{(j)} \in a]Q^{(j)}(Z^{(j)} \in a|Z^{(j')} \in y)} \\
&\geq \sum_{a=1}^{C}\sum_{y=1}^{C}\Pr\left[Z^{(j)} \in a, Z^{(j')} \in y\right]\log\frac{\Pr\left[Z^{(j)} \in a|Z^{(j')} \in y\right]}{Q^{(j)}(Z^{(j)} \in a|Z^{(j')} \in y)} \\
&\quad + \sum_{a=1}^{C}\sum_{y=1}^{C}\Pr\left[Z^{(j)} \in a, Z^{(j')} \in y\right]\log\frac{Q^{(j)}(Z^{(j)} \in a|Z^{(j')} \in y)}{\Pr[Z^{(j)} \in a]} \\
&= \mathrm{KL}\left(P(Z^{(j)}|Z^{(j')})\big\|Q^{(j)}(Z^{(j)}|Z^{(j')})\right) \\
&\quad + \sum_{a=1}^{C}\sum_{y=1}^{C}\Pr\left[Z^{(j)} \in a, Z^{(j')} \in y\right]\log\frac{Q^{(j)}(Z^{(j)} \in a|Z^{(j')} \in y)}{\Pr[Z^{(j)} \in a]} \\
&\geq \sum_{a=1}^{C}\sum_{y=1}^{C}\Pr\left[Z^{(j)} \in a, Z^{(j')} \in y\right]\log\frac{Q^{(j)}(Z^{(j)} \in a|Z^{(j')} \in y)}{\Pr[Z^{(j)} \in a]} \\
&= \sum_{a=1}^{C}\sum_{y=1}^{C}\Pr\left[Z^{(j)} \in a, Z^{(j')} \in y\right]\log Q^{(j)}(Z^{(j)} \in a|Z^{(j')} \in y) + H\left(P(Z^{(j)})\right) \\
&\geq \sum_{a=1}^{C}\sum_{y=1}^{C}\Pr\left[Z^{(j)} \in a, Z^{(j')} \in y\right]\log Q^{(j)}(Z^{(j)} \in a|Z^{(j')} \in y) \\
&\geq \frac{1}{n}\sum_{i=1}^{n}\sum_{a=1}^{C}\sum_{y=1}^{C}\phi\left(Z_i^{(j)} \in a, Z_i^{(j')} \in y\right)\log Q^{(j)}(Z^{(j)} \in a|Z^{(j')} \in y). \tag{9}
\end{aligned}
$$

$\square$

*Proof of Theorem 3.1.* The upper bound in Theorem 3.1 of the main paper follows from IB $= I(Z^{(j)}, X^{(j)}) - I(Z^{(j)}, Z^{(j')})$, the upper bound for $I(Z^{(j)}, X^{(j)})$ in Lemma C.1 and the lower bound for $I(Z^{(j)}, Z^{(j')})$ in Lemma C.2. $\square$

## C.1. Computation of $Q^{(j)}(Z^{(j)}|Z^{(j')})$

The variational distribution $Q^{(j)}(Z^{(j)}|Z^{(j')})$ is computed by

$$Q^{(j)}(Z^{(j)} \in a|Z^{(j')} \in y) = \Pr\left[Z^{(j)} \in a|Z^{(j')} \in y\right] = \frac{\sum_{i=1}^{n} \phi(Z_i^{(j)}, a)\mathbb{1}_{\left\{Z_i^{(j')} \in y\right\}}}{\sum_{i=1}^{n} \mathbb{1}_{\left\{Z_i^{(j')} \in y\right\}}}, \quad a, y \in [C]. \tag{10}$$

Our Algorithm 3 computes $Q^{(j)}(Z^{(j)}|Z^{(j')})$ efficiently with a time complexity of $\Theta(nC + nT_0)$, where $C$ is the number of classes, $n$ is the number of training samples, and $T_0$ denotes the computational complexity of a forward and backward pass of the neural network with respect to each training sample.

We remark that the variational distribution $Q^{(j)}(Z^{(j)} \in a|Z^{(j')} \in y) = \Pr\left[Z^{(j)} \in a|Y = y\right]$ is modeled through the conditional probability on the class label $Y = y$. We first emphasize that our upper bound IBB$^{(j)}$ in Theorem 3.1 for the IB loss IB$^{(j)}$ holds for an arbitrary distribution $Q^{(j)}(Z^{(j)} \in a|Z^{(j')} \in y)$. Our particular choice for $Q^{(j)}$ is not only effective in empirical performance as evidenced in Section 4, but also efficient which eliminates of the need of a time consuming process of training a VAE to model the conditional distribution $P(Z^{(j)}|Z^{(j')})$, such as CLUB (Cheng et al., 2020).

## C.2. Computation of IBB$^{(\text{fuse})}$

Repeating the argument in the proof of Theorem 3.1, the upper bound for the IB loss on the fused multimodal representation is IB$^{(\text{fuse})} \leq$ IBB$^{(\text{fuse})}$, where

$$\begin{aligned}
\text{IBB}^{(\text{fuse})} = &\frac{1}{n}\sum_{i=1}^{n}\sum_{a=1}^{C}\sum_{y=1}^{C} \phi\left(Z_i \in a, X_i^{(j)} \in y\right) \log\left(\frac{\phi\left(Z_i \in a, X_i^{(1)} \in y\right)}{p_y\phi(Z_i, a)}\right) \\
&+ \frac{1}{n}\sum_{i=1}^{n}\sum_{a=1}^{C}\sum_{y=1}^{C} \phi\left(Z_i \in a, X_i^{(2)} \in y\right) \log\left(\frac{\phi\left(Z_i \in a, X_i^{(2)} \in y\right)}{p_y\phi(Z_i, a)}\right) \\
&- \frac{1}{n}\sum_{i=1}^{n}\sum_{a=1}^{C}\sum_{y=1}^{C'} \phi(Z_i, a)\mathbb{1}_{\{Y_i=y\}} \log Q(Z \in a|Y = y),
\end{aligned} \tag{11}$$

where the variational distribution $Q(Z \in a|Y = y)$ is computed by

$$Q(Z \in a|Y = y) = \Pr\left[Z \in a|Y = y\right] = \frac{\sum_{i=1}^{n} \phi(Z_i, a)\mathbb{1}_{\{Y_i=y\}}}{\sum_{i=1}^{n} \mathbb{1}_{\{Y_i=y\}}}, \quad a \in [C], y \in [C']. \tag{12}$$

Our Algorithm 4 computes $Q(Z|Y)$ efficiently with a time complexity of $\Theta(nC + nT_0)$, where $C$ is the number of prototypes, $n$ is the number of training samples, and $T_0$ denotes the computational complexity of a forward and backward pass of the neural network with respect to each training sample.

## D. Additional Experiment Settings

### D.1. Datasets

For speech-based emotion recognition, we adopt CREMA-D (Cao et al., 2014), which consists of acted audio-visual recordings where professional actors convey six categorical emotions through synchronized facial expressions and speech

signals. In addition, we include IEMOCAP (Busso et al., 2008), a widely used conversational emotion corpus featuring dyadic interactions with fine-grained emotional annotations, allowing us to further assess robustness under spontaneous and context-dependent emotional expressions. For multimodal sentiment analysis (MSA), experiments are conducted on CMU-MOSI (Zadeh et al., 2016), which integrates language, acoustic, and visual modalities and provides real-valued sentiment intensity labels ranging from -3 (strongly negative) to +3 (strongly positive). We further extend sentiment evaluation to MELD (Poria et al., 2019), a large-scale multimodal dialogue dataset derived from television series, where utterance-level emotion and sentiment labels are annotated within multi-speaker conversational contexts. To study multimodal learning in biomedical settings, we consider eight pathological tissue datasets constructed from healthy human breast tissues (10x-hNB-A–H) and human breast cancer tissues (10x-hBC-A-D) (Xu et al., 2024). Each dataset jointly contains gene expression profiles and corresponding histology images. Following prior work (Mai et al., 2023a; Wu et al., 2025), IBMA is trained exclusively on healthy tissue datasets and subsequently transferred to cancer tissue datasets for pathological tissue detection, enabling evaluation under a cross-domain generalization setting. In addition, we involve large-scale multimodal learning datasets, MIMIC-CXR (Johnson et al., 2019) and CheXpert (Irvin et al., 2019), which are large-scale chest radiography datasets consisting of paired medical images and clinical text reports, and are widely used benchmarks for multimodal representation learning and medical image understanding. UPMC Food-101 (Wang et al., 2015) contains 101 food categories, while WIKI-DOC (Fujinuma et al., 2023) consists of 111 document classes. PME4 (Chen et al., 2022) is a four-modality emotion recognition benchmark containing audio, video, EEG, and EMG signals (Chen et al., 2022).

### D.2. Experiment Settings

IBMA employs modality-specific encoders and a cross-modal attention–based fusion architecture for multimodal learning across tasks, where each encoder produces a $512$ dimensional embedding as specified in the corresponding experimental sections. The cross-modal fusion module is implemented using a single transformer block (Vaswani et al., 2017) with 8 attention heads and a position-wise MLP, together with residual connections and layer normalization, after which the fused features are projected to a $512$ dimensional multimodal representation following (Wu et al., 2025). All models are trained for 500 epochs using the Adam optimizer, with the learning rate selected from $\{1 \times 10^{-4}, 5 \times 10^{-4}, 1 \times 10^{-3}\}$ and the weight decay from $\{1 \times 10^{-6}, 1 \times 10^{-5}, 1 \times 10^{-4}\}$. The balancing factor $\eta$ for the IBB regularization term is selected via five-fold cross-validation from $\{0.1, 0.2, \ldots, 0.9\}$ on the training set of each evaluation benchmark. The training of IBMA starts with a 5-epoch warm-up stage, during which only the cross-entropy loss is optimized, to ensure that the latent features generated after the warm-up stage are reasonable and informative. The centroids are computed as the average of the features after the warmup. In this way, the prototypes are already well and implicitly regularized as averaged features generated by a moderately trained neural network after the warmup, so there is no need for normalizing the centroids, and the norms of the centroids are free of the scaling problem in all the experiments in this paper. Such a strategy is also widely used in the literature (Snell et al., 2017; Yang et al., 2018).

### D.3. Compared Methods

We compare IBMA with a broad range of multimodal fusion and information-theoretic baselines, including early fusion method, Concat (Perez et al., 2018) and BiGated (Kiela et al., 2018), representation disentanglement method, MISA (Hazarika et al., 2020), and information bottleneck–based methods, Deep IB (Wang et al., 2019), MMIB (Zhang et al., 2022), MMRLIB (Cui et al., 2024), MIB (Mai et al., 2023a), MCIB (Wang et al., 2026), and OMIB (Wu et al., 2025), and contrastive learning based alignment methods, DCLF (Xie et al., 2025) and CLFA (Zhang et al., 2024a). The best results among the three variants of MIB (Mai et al., 2023a), including E-MIB, L-MIB, and C-MIB, are reported.

## E. Additional Experiment Results

### E.1. Multimodal Sentiment Analysis

We evaluate IBMA on multimodal sentiment analysis, where sentiment polarity is jointly inferred from heterogeneous modalities, following standard protocols in the multimodal learning literature (Cui et al., 2024; Mai et al., 2023a; Wu et al., 2025). For both CMU-MOSI and MELD, we adopt a tri-modal setting following (Wu et al., 2025), incorporating audio, visual, and textual modalities. Raw speech waveforms are transformed into log-Mel spectrograms and encoded with a ResNet-18 backbone, while facial video frames are processed by a shared-weight ResNet-18 encoder with temporal average pooling. Textual inputs are encoded using a BERT-based text encoder (Liu et al., 2019). The resulting modality-specific representations are fused through a cross-modal attention module to obtain a unified multimodal representation for

*Table 6.* Performance comparison for sentiment analysis on CMU-MOSI and MELD. The IBMA results are averaged over 10 runs with different random initializations and reported as mean $\pm$ standard deviation, while the statistical significance of its improvements over the best baseline is presented in Table 18 in Section E.9 of the appendix.

| Method | CMU-MOSI | | | MELD | |
|---|---|---|---|---|---|
| | Acc-7 | Acc-2 | F1 | Acc-2 | F1 |
| Concat (Perez et al., 2018) | 41.5 | 81.1 | 82.0 | 74.2 | 73.8 |
| BiGated (Kiela et al., 2018) | 41.8 | 82.1 | 83.2 | 75.1 | 74.6 |
| MISA (Hazarika et al., 2020) | 42.3 | 83.4 | 83.6 | 76.3 | 75.8 |
| Deep IB (Wang et al., 2019) | 45.3 | 83.2 | 83.3 | 76.0 | 75.5 |
| MMIB (Zhang et al., 2022) | 46.3 | 85.0 | 85.0 | 78.4 | 78.0 |
| MMRLIB (Cui et al., 2024) | 45.7 | 84.3 | 84.4 | 77.8 | 77.3 |
| DMIB (Fang et al., 2024) | 40.4 | 83.2 | 83.3 | 75.6 | 75.2 |
| MIB (Mai et al., 2023a) | 48.6 | 85.3 | 85.3 | 79.1 | 78.7 |
| MCIB (Wang et al., 2026) | 48.8 | 86.7 | 87.2 | 80.2 | 79.7 |
| CLFA (Zhang et al., 2024a) | 47.8 | 86.1 | 86.6 | 80.2 | 79.7 |
| DCLF (Xie et al., 2025) | 48.5 | 86.5 | 87.1 | 80.1 | 79.5 |
| OMIB (Wu et al., 2025) | 48.6 | 86.9 | 87.1 | 80.5 | 80.1 |
| **IBMA (Ours)** | **50.1**±0.2 | **87.9**±0.3 | **88.3**±0.2 | **82.0**±0.4 | **81.5**±0.3 |

sentiment prediction. We report seven-class accuracy (Acc-7) on CMU-MOSI to evaluate fine-grained sentiment polarity and intensity, and additionally report binary accuracy (Acc-2) and F1-score by grouping sentiments into positive and negative classes. For MELD, we report Acc-2 and F1-score only, as it provides binary sentiment annotations. As shown in Table 6, IBMA consistently outperforms existing multimodal learning methods on both benchmarks. In particular, IBMA surpasses the strongest baseline by 1.4% in F1-score and 1.5% in binary accuracy on MELD, demonstrating the effectiveness of modality-specific representation alignment.

### E.2. Anomalous Tissue Detection

We further evaluate IBMA on multimodal anomalous tissue detection, where the objective is to identify anomalous tissue regions from eight human breast cancer datasets (10x-hBC-A–D) by jointly modeling gene expression and histology modalities. Due to the limited availability of region-level anomaly annotations, we adopt the Support Vector Data Description (SVDD) framework (Ruff et al., 2018) for anomaly detection. Following OMIB (Wu et al., 2025), we employ modality-specific encoders, where gene expression profiles are encoded using a multi-layer perceptron (MLP) and histology regions are encoded using a ResNet-18 encoder. The resulting modality-specific representations are fused using the cross-modal attention module, leading to the fused multimodal representation. The same modality-specific encoders are used across all baseline methods to ensure a fair comparison with IBMA. The model is trained exclusively on eight healthy tissue datasets (10x-hNB-A–H) to learn a compact hypersphere that encloses multimodal representations of normal tissue regions. During inference, the trained model is applied to each breast cancer dataset, and the distance between each multimodal representation and the hypersphere center is used as an anomaly score to identify anomalous regions. All competing methods follow the same SVDD-based training and evaluation protocol and use identical modality-specific encoders for fair comparison. Detection performance is evaluated using ROC-AUC and F1-score, computed based on the predicted anomaly scores and available region-level annotations. It is observed in Table 7 that IBMA significantly outperforms the competing multimodal learning methods on anomalous tissue detection. For instance, IBMA outperforms the strongest baseline, OMIB, by 2% in AUC averaged on all four test datasets.

*Table 7.* Performance comparison of multimodal learning methods for anomalous tissue detection on the 10x-hBC-{A–D} datasets. The IBMA results are averaged over 10 runs with different random initializations and reported as mean $\pm$ standard deviation, while the statistical significance of its improvements over the best baseline is presented in Table 20 in Section E.9 of the appendix.

| Method | 10x-hBC-A | | 10x-hBC-B | | 10x-hBC-C | | 10x-hBC-D | | Mean | |
|---|---|---|---|---|---|---|---|---|---|---|
| | AUC | F1 | AUC | F1 | AUC | F1 | AUC | F1 | AUC | F1 |
| Concat (Perez et al., 2018) | 0.537 | 0.884 | 0.866 | 0.654 | 0.638 | 0.750 | 0.555 | 0.509 | 0.649 | 0.699 |
| BiGated (Kiela et al., 2018) | 0.489 | 0.821 | 0.518 | 0.352 | 0.563 | 0.727 | 0.540 | 0.494 | 0.528 | 0.599 |
| MISA (Hazarika et al., 2020) | 0.498 | 0.873 | 0.499 | 0.213 | 0.586 | 0.754 | 0.495 | 0.450 | 0.520 | 0.573 |
| Deep IB (Wang et al., 2019) | 0.522 | 0.878 | 0.379 | 0.102 | 0.433 | 0.693 | 0.484 | 0.443 | 0.455 | 0.529 |
| MMIB (Zhang et al., 2022) | 0.623 | 0.894 | 0.818 | 0.559 | 0.765 | 0.822 | 0.501 | 0.465 | 0.677 | 0.685 |
| MMRLIB (Cui et al., 2024) | 0.626 | 0.897 | 0.817 | 0.583 | 0.662 | 0.783 | 0.604 | 0.524 | 0.677 | 0.697 |
| DMIB (Fang et al., 2024) | 0.423 | 0.865 | 0.849 | 0.607 | 0.743 | 0.827 | 0.642 | 0.540 | 0.664 | 0.710 |
| MIB (Mai et al., 2023a) | 0.598 | 0.891 | 0.770 | 0.483 | 0.659 | 0.786 | 0.652 | 0.564 | 0.602 | 0.681 |
| MCIB (Wang et al., 2026) | 0.684 | 0.898 | 0.852 | 0.628 | 0.716 | 0.811 | 0.621 | 0.548 | 0.718 | 0.721 |
| CLFA (Zhang et al., 2024a) | 0.663 | 0.892 | 0.834 | 0.612 | 0.704 | 0.803 | 0.610 | 0.541 | 0.703 | 0.712 |
| DCLF (Xie et al., 2025) | 0.701 | 0.900 | 0.872 | 0.641 | 0.728 | 0.819 | 0.632 | 0.552 | 0.733 | 0.728 |
| OMIB (Wu et al., 2025) | 0.728 | 0.904 | 0.903 | 0.663 | 0.743 | 0.820 | 0.640 | 0.561 | 0.754 | 0.737 |
| **IBMA (Ours)** | **0.744**±0.003 | **0.915**±0.002 | **0.916**±0.002 | **0.685**±0.005 | **0.767**±0.004 | **0.840**±0.003 | **0.668**±0.004 | **0.577**±0.004 | **0.774** | **0.754** |

### E.3. Additional Experiments on PME4 with Four Modalities

We further evaluate IBMA on PME4 (Chen et al., 2022), a multimodal benchmark containing four modalities, including audio, video, EEG, and EMG, in order to assess the effectiveness of IBMA in a more challenging multimodal setting. For PME4, we employ a pretrained CNN-based encoder for visual inputs, a pretrained audio encoder for speech signals, and lightweight neural encoders for EEG and EMG signals. The resulting modality-specific representations are fused by the same cross-modal attention module used in the other experiments, while modality-specific representation alignment is applied across modalities during training.

*Table 8.* Accuracy comparison on PME4.

| Method | PME4 Accuracy (%) |
|---|---|
| Concat (Perez et al., 2018) | 78.6 |
| BiGated (Kiela et al., 2018) | 79.4 |
| MISA (Hazarika et al., 2020) | 79.1 |
| Deep IB (Wang et al., 2019) | 78.9 |
| MMIB (Zhang et al., 2022) | 79.8 |
| MMRLIB (Cui et al., 2024) | 80.0 |
| MIB (Mai et al., 2023a) | 80.6 |
| MCIB (Wang et al., 2026) | 80.8 |
| CLFA (Zhang et al., 2024a) | 80.7 |
| DCLF (Xie et al., 2025) | 80.9 |
| OMIB (Wu et al., 2025) | 81.1 |
| **IBMA (Ours)** | **82.2** |

As shown in Table 8, IBMA significantly outperforms all competing multimodal learning methods on PME4. In particular, IBMA achieves an accuracy of $82.2\%$, outperforming the strongest baseline, OMIB (Wu et al., 2025), by $1.1\%$. These results further demonstrate the effectiveness of IBMA in learning informative and semantically aligned representations in multimodal settings with more than two modalities. We also report the training time comparison between IBMA and OMIB in Table 9. It is observed that IBMA achieves the improved classification performance with only a modest increase in training time.

*Table 9.* Training time comparison between OMIB and IBMA on PME4.

| Method | PME4 Accuracy (%) | Training Time (Min/Epoch) |
|---|---|---|
| OMIB (Wu et al., 2025) | 81.1 | 4.3 |
| **IBMA (Ours)** | **82.2** | **4.6** |

### E.4. Comparison with Other Modality-Specific Regularization Methods

We further compare IBMA with two ablation models that replace the modality-specific IB alignment loss with other modality-specific regularization methods for cross-modal feature alignment. In particular, we consider a contrastive ablation model that employs the CLIP-based contrastive loss (Zhang et al., 2024b) and a distillation ablation model that employs the distillation loss (Li et al., 2023) for modality-specific alignment. All the other components, including the modality-specific encoders, the cross-modal fusion module, and the training settings, are kept the same as those in IBMA. As shown in Table 10, IBMA significantly outperforms both ablation models on MIMIC-CXR and CheXpert. For example, IBMA outperforms the contrastive ablation model by $1.9\%$ on MIMIC-CXR and $1.8\%$ on CheXpert, which demonstrates the advantage of the proposed modality-specific IB alignment over existing modality-specific regularization methods for multimodal learning.

*Table 10.* Comparison with other modality-specific regularization methods for multimodal disease classification.

| Methods | MIMIC-CXR | CheXpert |
|---|---|---|
| Contrastive Ablation Model | 70.8 | 89.3 |
| Distillation Ablation Model | 71.0 | 89.2 |
| **IBMA (Ours)** | **72.7** | **91.1** |

### E.5. Additional Experiments on Datasets with More Classes

We further evaluate IBMA on two multimodal datasets with a larger number of classes, namely UPMC Food-101 (Wang et al., 2015) and WIKI-DOC (Fujinuma et al., 2023), to study the scalability of IBMA to settings with large class cardinality.

UPMC Food-101 contains 101 food categories, while WIKI-DOC consists of 111 document classes. Following the standard multimodal classification setting, we employ pretrained ViT-B and BERT-base as the image and text encoders, respectively. The resulting modality-specific representations are fused by the same cross-modal attention module used in the other experiments.

*Table 11.* Accuracy (%) comparison on multimodal datasets with more classes.

| Method | UMPC Food-101 | WIKI-DOC |
|---|---|---|
| Class Number | 101 | 111 |
| Concat (Perez et al., 2018) | 89.4 | 91.3 |
| BiGated (Kiela et al., 2018) | 90.1 | 91.9 |
| MISA (Hazarika et al., 2020) | 89.9 | 91.7 |
| Deep IB (Wang et al., 2019) | 89.6 | 91.5 |
| MMIB (Zhang et al., 2022) | 90.2 | 92.1 |
| MMRLIB (Cui et al., 2024) | 90.3 | 92.3 |
| MIB (Mai et al., 2023a) | 90.8 | 92.6 |
| MCIB (Wang et al., 2026) | 91.0 | 92.7 |
| CLFA (Zhang et al., 2024a) | 90.9 | 92.6 |
| DCLF (Xie et al., 2025) | 91.1 | 92.8 |
| OMIB (Wu et al., 2025) | 91.2 | 93.0 |
| **IBMA (Ours)** | **92.4** | **94.1** |

As shown in Table 11, IBMA consistently outperforms all competing multimodal learning methods on both datasets. In particular, IBMA achieves 92.4% accuracy on UPMC Food-101 and 94.1% accuracy on WIKI-DOC, outperforming the strongest baseline, OMIB, by 1.2% and 1.1%, respectively. These results demonstrate that IBMA remains effective when the number of classes becomes substantially larger. We also report the training time comparison between IBMA and OMIB in Table 12. It is observed that IBMA achieves improved classification performance with only a modest increase in training time.

*Table 12.* Training time comparison between OMIB and IBMA on datasets with more classes.

| Dataset | Training Time (Min/Epoch) | |
|---|---|---|
| | OMIB | IBMA |
| UPMC Food-101 | 6.8 | 7.4 |
| WIKI-DOC | 7.6 | 8.1 |

## E.6. Sensitivity Analysis

We analyze the sensitivity of IBMA to the balancing factor $\eta$ in the joint training loss $\mathcal{L}_b$ from Equation (1) on CREMA-D, where $\eta$ controls the strength of the IBB regularization relative to the cross-entropy loss. We conduct the sensitivity study by varying $\eta$ over the range $\{0.1, 0.2, \ldots, 0.9\}$ while keeping all other training settings fixed. As shown in Table 13, IBMA maintains stable performance across different values of $\eta$, with an accuracy variation of at most 0.4% on CREMA-D for multimodal emotion recognition.

*Table 13.* Sensitivity of IBMA to the balancing coefficient $\eta$ on CREMA-D for multimodal emotion recognition.

| $\eta$ | 0.1 | 0.2 | 0.3 | 0.4 | 0.5 | 0.6 | 0.7 | 0.8 | 0.9 |
|---|---|---|---|---|---|---|---|---|---|
| Accuracy (%) | 65.1 | 65.0 | 65.2 | 65.3 | **65.4** | 65.3 | 65.1 | 65.0 | 65.0 |

In addition, we study the sensitivity of IBMA to the number of prototypes used for prototype-based feature distribution modeling. Specifically, we vary the number of prototypes $C$ from $2 \times C'$ to $20 \times C'$ on MELD, where $C$ denotes the number of prototypes and $C'$ denotes the number of classes, while keeping all other training settings fixed. As shown in Table 14, the performance of IBMA remains stable across different choices of the number of prototypes, with only minor variations in accuracy. These results further demonstrate the robustness of IBMA with respect to the number of prototypes.

*Table 14.* Sensitivity of IBMA to the number of prototypes on MELD. Here $C$ denotes the number of prototypes and $C'$ denotes the number of classes.

| $C$ | $2 \times C'$ | $5 \times C'$ | $10 \times C'$ | $15 \times C'$ | $20 \times C'$ |
|---|---|---|---|---|---|
| Accuracy (%) | 66.0 | **66.3** | **66.3** | 66.2 | 66.2 |

### E.7. Extension to Self-Supervised Multimodal Pretraining

Although IBMA is presented in the supervised multimodal classification setting, its prototype-based formulation can be naturally extended to self-supervised multimodal pretraining. We evaluate IBMA on UPMC Food-101 and WIKI-DOC under the self-supervised multimodal pretraining setting, and report linear evaluation accuracy after pretraining. We compare IBMA with CLIP-style contrastive pretraining (Radford et al., 2021), SimCLR-style multimodal pretraining (Chen et al., 2020), and a self-supervised prototype-based variant of OMIB. As shown in Table 15, IBMA consistently outperforms all competing methods on both datasets. In particular, IBMA improves the linear evaluation accuracy over OMIB-PL by $0.9\%$ on UPMC Food-101 and $0.9\%$ on WIKI-DOC. These results suggest that the proposed prototype-based modality-specific IB alignment is not restricted to supervised learning and can also benefit self-supervised multimodal pretraining.

*Table 15.* Linear evaluation accuracy comparison under self-supervised multimodal pretraining.

| Method | UPMC Food-101 Linear Eval Acc (%) | WIKI-DOC Linear Eval Acc (%) |
|---|---|---|
| CLIP-style Contrastive Pretraining | 88.7 | 90.8 |
| SimCLR-style Multimodal Pretraining | 88.1 | 90.2 |
| OMIB-PL (Self-Supervised) | 89.5 | 91.4 |
| **IBMA (Self-Supervised)** | **90.4** | **92.3** |

### E.8. Training Time Comparison

We further compare the training efficiency of IBMA with existing multimodal learning methods on the CREMA-D dataset. All methods are trained on a single NVIDIA A100 GPU. The training time results on CREMA-D are summarized in Table 16, where the total training time for both IBMA and the baseline methods is reported. It is observed in Table 16 that IBMA only marginally increases the training time by $5.3\%$ compared to the strongest baseline OMIB, while significantly outperforming OMIB by $1.8\%$ in emotion recognition accuracy on CREMA-D, demonstrating the effectiveness and efficiency of the proposed IBMA. It is worthwhile to mention that the prototypes in the input feature space can be efficiently pre-computed prior to training using FAISS (Douze et al., 2025), which supports fast, memory-efficient, and highly parallelized centroid and distance computations on GPUs, even for high-dimensional features. In particular, computing the prototypes and the distances between the input features of all samples in the training set of CREMA-D and their corresponding prototypes requires only 16.7 seconds, which happens one time before the training. Computing the prototypes and the distances between the learned features of all samples in the training set of CREMA-D and their corresponding prototypes requires only 2.1 seconds in each epoch.

*Table 16.* Performance and training time comparison of multimodal learning methods on CREMA-D.

| Method | CREMA-D | |
|---|---|---|
| | Accuracy (%) | Training Time (Minutes/Epoch) |
| Concat (Perez et al., 2018) | 53.2 | 1.5 |
| BiGated (Kiela et al., 2018) | 58.4 | 1.7 |
| MISA (Hazarika et al., 2020) | 57.7 | 1.8 |
| Deep IB (Wang et al., 2019) | 54.1 | 1.8 |
| MMIB (Zhang et al., 2022) | 56.7 | 1.9 |
| MMRLIB (Cui et al., 2024) | 57.3 | 2.2 |
| MIB (Mai et al., 2023a) | 61.4 | 1.8 |
| MCIB (Wang et al., 2026) | 63.2 | 1.9 |
| CLFA (Zhang et al., 2024a) | 63.0 | 2.0 |
| DCLF (Xie et al., 2025) | 63.4 | 2.0 |
| OMIB (Wu et al., 2025) | 63.6 | 1.9 |
| **IBMA (Ours)** | **65.4**±0.4 | 2.0 |

In addition, we analyze the sensitivity of the performance of IBMA to the prototype number $C$. We conduct the

### E.9. Improvement Significance Analysis

To verify that the performance improvements achieved by IBMA over existing multimodal learning methods are statistically significant and not caused by random variation, we conduct paired $t$-tests between IBMA and the best-performing baseline method on each dataset. For all the multimodal learning tasks, IBMA and the corresponding best baseline are trained and evaluated 10 times with different random seeds for network initialization. The mean and standard deviation of the evaluation metrics, together with the resulting $p$-values of the paired $t$-tests, are reported in Table 17 for emotion recognition, Table 18 for sentiment analysis, Table 19 for multimodal disease classification, and Table 20 for anomalous

tissue detection. As shown in Tables 17–20, the largest $p$-value across all benchmarks and tasks is $3.6 \times 10^{-5}$ on MELD for emotion recognition, which is below the standard significance threshold of 0.05. These results indicate that the performance improvements achieved by IBMA over the corresponding best baseline methods are statistically significant with $p \ll 0.05$, and are unlikely to be caused by random variation.

*Table 17.* Statistical significance analysis for multimodal emotion recognition. Paired $t$-tests are conducted between IBMA and the best baseline on each dataset over 10 runs.

| Dataset | Best Baseline (Accuracy) | IBMA (Accuracy) | $p$-value |
|---|---|---|---|
| CREMA-D | $63.6 \pm 0.3$ | $65.4 \pm 0.4$ | $1.2 \times 10^{-6}$ |
| MELD | $64.3 \pm 0.4$ | $66.3 \pm 0.3$ | $3.6 \times 10^{-5}$ |
| IEMOCAP | $74.3 \pm 0.3$ | $75.7 \pm 0.3$ | $4.5 \times 10^{-6}$ |

*Table 18.* Statistical significance analysis for multimodal sentiment analysis. Paired $t$-tests are conducted between IBMA and the best baseline on each dataset over 10 runs.

| Dataset | Best Baseline (Acc-2) | IBMA (Acc-2) | $p$-value |
|---|---|---|---|
| CMU-MOSI | $86.9 \pm 0.3$ | $87.9 \pm 0.3$ | $2.2 \times 10^{-7}$ |
| MELD | $80.5 \pm 0.4$ | $82.0 \pm 0.4$ | $1.9 \times 10^{-6}$ |

*Table 19.* Statistical significance analysis for multimodal disease classification. Paired $t$-tests are conducted between IBMA and the best baseline on each dataset over 10 runs.

| Dataset | Best Baseline (mAUC) | IBMA (mAUC) | $p$-value |
|---|---|---|---|
| MIMIC-CXR | $71.0 \pm 0.2$ | $72.7 \pm 0.3$ | $3.6 \times 10^{-6}$ |
| CheXpert | $89.3 \pm 0.2$ | $91.1 \pm 0.2$ | $1.1 \times 10^{-6}$ |

*Table 20.* Statistical significance analysis for multimodal anomalous tissue detection. Paired $t$-tests are conducted between IBMA and the best baseline over 10 runs.

| Dataset | Best Baseline (AUC) | IBMA (AUC) | $p$-value |
|---|---|---|---|
| 10x-hBC-A | $0.728 \pm 0.004$ | $0.744 \pm 0.003$ | $2.5 \times 10^{-7}$ |
| 10x-hBC-B | $0.903 \pm 0.002$ | $0.916 \pm 0.002$ | $1.6 \times 10^{-6}$ |
| 10x-hBC-C | $0.743 \pm 0.003$ | $0.767 \pm 0.004$ | $5.7 \times 10^{-7}$ |
| 10x-hBC-D | $0.652 \pm 0.004$ | $0.668 \pm 0.004$ | $4.2 \times 10^{-7}$ |

To further verify that the performance improvements brought by IBB over existing IB loss upper bounds are statistically significant rather than due to random variation, we additionally conduct paired $t$-tests between the IBB-based model and each competing variant. All models are trained and evaluated 10 times on CREMA-D with different random seeds under identical experimental settings. As shown in Table 21, the performance gains achieved by IBB over VIB, APIB, and CLUB are all statistically significant, with $p$-values well below the standard significance threshold of 0.05. These results further confirm that the superiority of IBB is consistent and robust, and cannot be attributed to stochastic training effects.

*Table 21.* Statistical significance analysis for comparing IBB with the best-performing IB loss upper bound on CREMA-D. Paired $t$-tests are conducted over 10 runs with different random seeds.

| Dataset | Best Baseline (Accuracy) | IBB (Accuracy) | $p$-value |
|---|---|---|---|
| CREMA-D | $64.1 \pm 0.3$ | $65.4 \pm 0.4$ | $2.8 \times 10^{-7}$ |

To further verify that the improvements brought by modality-specific representation alignment are statistically significant rather than caused by random variation, we additionally conduct paired $t$-tests between IBMA and the ablation model without modality-specific representation alignment. Both models are trained and evaluated 10 times with different random seeds under identical experimental settings. The resulting $p$-values for multimodal disease classification are reported in Table 22.

### E.10. Analysis on the Modality-Specific Encoder

To explicitly assess the quality of modality-specific representations learned by IBMA, we evaluate each modality-specific encoder independently on the CheXpert dataset. Following the standard evaluation protocols in existing works (Mai et al.,

*Table 22.* Statistical significance analysis comparing IBMA with the ablation model without modality-specific representation alignment. Paired $t$-tests are conducted over 10 runs with different random seeds.

| Dataset | IBMA w/o Modality-Specific Representation Alignment (mAUC) | IBMA (mAUC) | $p$-value |
|---|---|---|---|
| MIMIC-CXR | $71.4 \pm 0.3$ | $72.7 \pm 0.3$ | $3.1 \times 10^{-6}$ |
| CheXpert | $89.7 \pm 0.2$ | $91.1 \pm 0.2$ | $1.4 \times 10^{-6}$ |

2023a), we attach a linear classifier to the frozen image encoder and text encoder, respectively, and train the classifier to predict thoracic disease labels using only a single modality at a time. We compare IBMA with OMIB (Wu et al., 2025) and an ablation variant, IBMA without modality-specific representation alignment, which removes all modality-specific IB losses in IBMA. It is observed in Table 23 that IBMA consistently improves the performance of both the image encoder and the text encoder compared to the competing methods. In particular, IBMA achieves significantly better mAUC than OMIB for both modalities, while removing modality-specific representation alignment leads to degraded unimodal performance. For instance, IBMA outperforms OMIB by $1.2\%$ and $1.5\%$ in mAUC for disease classification on the image domain and text domain, eventually leading to an improvement of $1.4\%$ in mAUC for multimodal disease classification. The consistent improvements across both modalities demonstrate that both modality-specific encoders benefit from the cross-domain supervision enforced by the modality-specific representation alignment proposed in IBMA.

*Table 23.* Evaluation of modality-specific encoders on CheXpert. Each encoder is evaluated independently using a linear classifier on top of frozen representations. Higher mAUC indicates better modality-specific representation quality.

| Method | mAUC | | |
|---|---|---|---|
| | Image | Text | Multimodal |
| OMIB (Wu et al., 2025) | 88.0 | 76.0 | 89.3 |
| IBMA w/o Modality-Specific Representation Alignment | 88.4 | 76.5 | 89.7 |
| **IBMA (Ours)** | **89.2** | **77.5** | **91.1** |

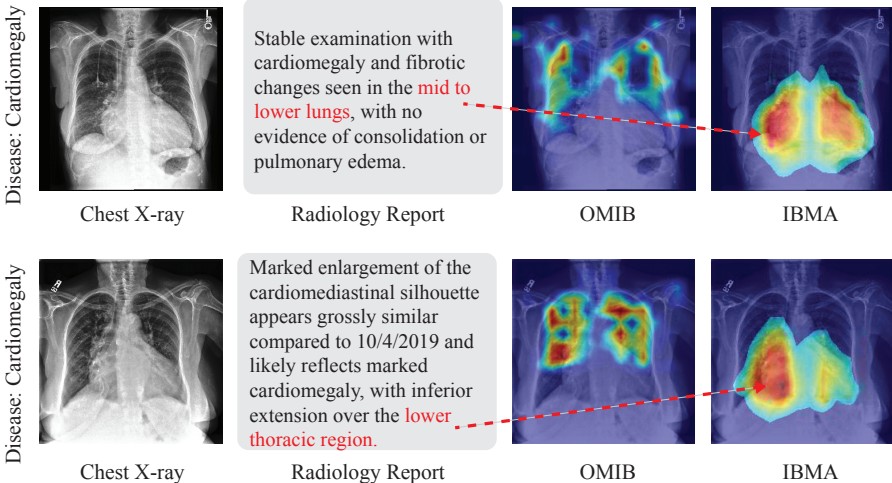

*Figure 3.* Grad-CAM visualization of the image encoder from OMIB and IBMA for two instances from the CheXpert (Irvin et al., 2019) dataset for multimodal disease classification. The two examples are in the class of cardiomegaly, which is characterized by abnormal enlargement of the heart, commonly reflected on chest X-rays as an increased cardiothoracic ratio observed in the lower middle of the two lungs. The radiology reports in both examples explicitly indicate the presence of cardiomegaly, using phrases such as "mid to lower lungs" and "lower thoracic region", which correspond to the typical anatomical location of cardiomegaly in chest radiographs. The red arrows illustrate that the image encoder of IBMA successfully focuses on the most informative regions of cardiomegaly in the mid to lower or lower lungs, as indicated in the radiology reports.

### E.11. Grad-CAM Visualization for Multimodal Disease Classification

As illustrated by the Grad-CAM visualization of the image encoders for multimodal disease classification in Figure 3, although the radiology reports in both examples provide clear location-specific indications of cardiomegaly, the state-of-the-art IB-based multimodal learning method, OMIB (Wu et al., 2025), fails to consistently attend to the most disease-relevant regions in the mid to lower lungs and instead focuses on the upper lung regions. In contrast, our proposed IBMA accurately

concentrates on the clinically meaningful regions associated with cardiomegaly as described in the radiology reports, which are the mid to lower regions in the lung, guided by the explicit textual cues. This is because the modality-specific representation alignment proposed in IBMA enables effective cross-modal textual guidance for image representations, which is absent in existing IB-based multimodal learning methods such as OMIB (Wu et al., 2025).

### E.12. t-SNE Visualization Analysis

To further investigate the representation quality learned by IBMA, we visualize the learned modality-specific and multimodal representations using t-SNE. Specifically, we compare the feature embeddings learned by IBMA and OMIB (Wu et al., 2025) on the same multimodal classification setting. As illustrated in Figure 4, the representations learned by IBMA form more compact and well-separated clusters than those learned by OMIB. In particular, IBMA exhibits less inter-class overlap and stronger intra-class consistency in both the modality-specific and fused multimodal feature spaces. These results provide qualitative evidence that the proposed modality-specific IB alignment enables IBMA to learn more discriminative and semantically aligned representations than OMIB.

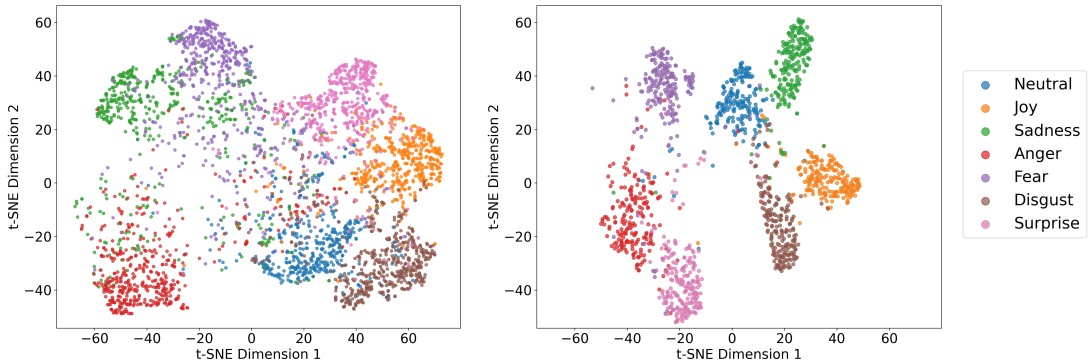

*Figure 4.* t-SNE visualization of the modality-specific and multimodal representations learned by OMIB and IBMA. Compared with OMIB, IBMA yields more compact and well-separated clusters in the feature space, with less inter-class overlap and better intra-class consistency.

---

**Algorithm 2** MI Minimization with vCLUB (Algorithm 1 in CLUB (Cheng et al., 2020))

---

1: **for** each training iteration **do**
2:     Sample $\{(X_i, Z_i)\}_{i=1}^{n}$ from $p_\sigma(X, Z)$
3:     Compute log-likelihood $\mathcal{L}(\theta) = \frac{1}{n}\sum_{i=1}^{n} \log q_\theta(Z_i|X_i)$
4:     Update $q_\theta(Z|X)$ by increasing $\mathcal{L}(\theta)$
5:     **for** $i = 1$ to $n$ **do**
6:         $L_i = \log q_\theta(Z_i|X_i) - \frac{1}{n}\sum_{j=1}^{n} \log q_\theta(Z_j|X_i)$
7:     **end for**
8:     Update $p_\sigma(X, Z)$ by reducing $\widehat{I}_{\text{vCLUB}} = \frac{1}{n}\sum_{i=1}^{n} L_i$
9: **end for**

---

## F. Computational Complexity Analysis

Given samples $\{(X_i, Z_i)\}_{i=1}^{n}$ drawn from the joint distribution $p_\sigma(X, Z) = p_\sigma(Z|X)p(X)$, the goal of CLUB is to optimize the parameters of the predictive neural network $p_\sigma(Z|X)$ such that the resulting joint model $p_\sigma(X, Z)$ induces minimal mutual information between $X$ and $Z$. Here, $p_\sigma(Z|X)$ serves as the main predictive model, such as an encoding model or a classification model, while $p(X)$ denotes the empirical data distribution. In addition, $q_\theta(Z|X)$ is a variational conditional distribution, also implemented as a neural network, used to approximate $p(Z|X)$ and to provide a differentiable estimator of mutual information. The optimization alternates between updating $q_\theta(Z|X)$ by increasing the log-likelihood $\mathcal{L}(\theta) = \frac{1}{n}\sum_{i=1}^{n} \log q_\theta(Z_i|X_i)$ to improve the approximation accuracy, and updating $p_\sigma(X, Z)$ by reducing the upper bound for the mutual information, $I(X, Z)$, $\widehat{I}_{\text{vCLUB}} = \frac{1}{n}\sum_{i=1}^{n} L_i$, where $L_i = \log q_\theta(Z_i|X_i) - \frac{1}{n}\sum_{j=1}^{n} \log q_\theta(Z_j|X_i)$. Through this alternating process, the algorithm jointly learns an accurate variational estimator $q_\theta$ and a predictive model $p_\sigma$. The complete training procedure is summarized in Algorithm 2.

---

**Algorithm 3** Efficient Computation of $Q^{(j)}(Z^{(j)} \in a \mid Z^{(j')} \in y)$

---

**Require:** The precomputed soft assignments $\phi(Z_i^{(j)}, a)$ for $i \in [n]$ and $a \in [C]$, the class labels $\{Y_i\}_{i=1}^n$, and the number of classes $C$. Here $\left\{Z_i^{(j)}\right\}_{i=1}^n$ are computed at the $t$-th epoch in Algorithm 1 of the main paper.
**Ensure:** Conditional distribution matrix $Q \in \mathbb{R}^{C \times C}$
 1: Initialize $Q \leftarrow \mathbf{0}^{C \times C}$ and count vector $M \leftarrow \mathbf{0}^C$.
 2: **for** $i = 1$ to $n$ **do**
 3:   **for** $a = 1$ to $C$ **do**
 4:     $Q[a, Y_i] \leftarrow Q[a, Y_i] + \phi(Z_i^{(j)}, a)$
 5:   **end for**
 6:   $M[Y_i] \leftarrow M[Y_i] + 1$
 7: **end for**
 8: **for** $y = 1$ to $C$ **do**
 9:   **for** $a = 1$ to $C$ **do**
10:     $Q[a, y] \leftarrow Q[a, y]/M[y]$
11:   **end for**
12: **end for**
13: **Return** $Q$

---

**Algorithm 4** Efficient Computation of $Q(Z \in a \mid Y = y)$

---

**Require:** The precomputed soft assignments $\phi(Z_i, a)$ for $i \in [n]$ and $a \in [C]$, the class labels $\{Y_i\}_{i=1}^n$, and the number of classes $C$. Here $\{Z_i\}_{i=1}^n$ are computed at the $t$-th epoch in Algorithm 1 of the main paper.
**Ensure:** Conditional distribution matrix $Q \in \mathbb{R}^{C \times C'}$
 1: Initialize $Q \leftarrow \mathbf{0}^{C \times C'}$ and count vector $M \leftarrow \mathbf{0}^{C'}$.
 2: **for** $i = 1$ to $n$ **do**
 3:   **for** $a = 1$ to $C$ **do**
 4:     $Q[a, Y_i] \leftarrow Q[a, Y_i] + \phi(Z_i, a)$
 5:   **end for**
 6:   $M[Y_i] \leftarrow M[Y_i] + 1$
 7: **end for**
 8: **for** $y = 1$ to $C'$ **do**
 9:   **for** $a = 1$ to $C$ **do**
10:     $Q[a, y] \leftarrow Q[a, y]/M[y]$
11:   **end for**
12: **end for**
13: **Return** $Q$

---

Suppose the computation of $q_\theta(Z|X)$ costs $T_q$ and that of $p_\sigma(Z|X)$ costs $T_p$. Computing $\mathcal{L}(\theta) = \frac{1}{n}\sum_{i=1}^n \log q_\theta(Z_i|X_i)$ requires $\Theta(nT_q)$ time. The computation of vCLUB $\widehat{I}_{\text{vCLUB}} = \frac{1}{n}\sum_{i=1}^n \left[ \log q_\theta(Z_i|X_i) - \frac{1}{n}\sum_{j=1}^n \log q_\theta(Z_j|X_i) \right]$ requires computing $[\log q_\theta(Z_j|X_i)]$ for $i, j \in [n]$, which requires $\Theta(n^2 T_q)$ time. Updating $p_\sigma$ for reducing $\widehat{I}_{\text{vCLUB}}$ over the same $n$ inputs adds $nT_p$ time. Hence, one training epoch requires $\Theta(n^2 T_q + nT_q + nT_p) = \Theta(n^2 T_q + nT_p)$.

**Computational Complexity of IBB$^{(j)}$.** Herein we analyze the computational complexity for computing the variational upper bound IBB$^{(j)}$ for the modality-specific IB loss. Let $T_0$ denote the complexity of a forward and backward computation of the model predicting $Z_i^{(j)}$ for one sample. For each epoch, the the computation of $\{Z_i^{(j)}\}_{i=1}^n$ using the modality-specific encoder takes $\Theta(nT_0)$ time. Given $\{Z_i^{(j)}\}_{i=1}^n$, we first pre-compute $\phi(Z_i^{(j)}, a)$ for $i \in [n]$ and $a \in [C]$, which takes $\Theta(nC)$ time.

To compute IBB$^{(j)} = \frac{1}{n}\sum_{i=1}^n \left(U_i^{(j)} - V_i^{(j)}\right)$, we separately compute $\frac{1}{n}\sum_{i=1}^n U_i^{(j)}$, which is the upper bound for $I(Z^{(j)}, X^{(j)})$, and $\frac{1}{n}\sum_{i=1}^n V_i^{(j)}$, which is the lower bound for $I(Z^{(j)}, Z^{(j')})$.

Let $A_{ia}^{(j)} := \exp(-\|Z_i^{(j)} - \mathcal{F}_a^{(j)}\|_2^2)$ and $B_{iy}^{(j)} := \exp(-\|X_i^{(j)} - \mathcal{C}_y^{(j)}\|_2^2)$ for all $a \in [C]$ and $y \in [C]$. Let $D_i^{(j)} := \sum_{a'=1}^{C} \sum_{y'=1}^{C} A_{ia'}^{(j)} B_{iy'}^{(j)} = \widehat{A}_i^{(j)} \widehat{B}_i^{(j)}$, where $\widehat{A}_i^{(j)} := \sum_{a'=1}^{C} A_{ia'}^{(j)}$ and $\widehat{B}_i^{(j)} := \sum_{y'=1}^{C} B_{iy'}^{(j)}$. For each $i \in [n]$, we first pre-compute all $A_{ia}^{(j)}$ and $B_{iy}^{(j)}$ for $a \in [C]$ and $y \in [C]$, which takes $\Theta(C)$ time. For each $i \in [n]$, we then pre-compute $\widehat{A}_i^{(j)}$, $\widehat{B}_i^{(j)}$, and $D_i^{(j)}$, which takes $\Theta(C)$ time. Then $U_i^{(j)}$ can be computed as

$$
\begin{aligned}
U_i^{(j)} &= \sum_{a=1}^{C} \sum_{y=1}^{C} \phi(Z_i^{(j)} \in a, X_i^{(j)} \in y) \log \left( \frac{\phi(Z_i^{(j)} \in a, X_i^{(j)} \in y)}{p_y \phi(Z_i^{(j)}, a)} \right) \\
&= \frac{1}{D_i^{(j)}} \sum_{a=1}^{C} \sum_{y=1}^{C} A_{ia}^{(j)} B_{iy}^{(j)} \log \left( \frac{A_{ia}^{(j)} B_{iy}^{(j)} / D_i^{(j)}}{p_y \phi(Z_i^{(j)}, a)} \right) \\
&= -\log D_i^{(j)} - \frac{\widehat{A}_i^{(j)}}{D_i^{(j)}} \sum_{y=1}^{C} B_{iy}^{(j)} \log p_y + \frac{1}{D_i^{(j)}} \left( \sum_{a=1}^{C} A_{ia}^{(j)} \log A_{ia}^{(j)} \right) \widehat{B}_i^{(j)} \\
&\quad + \frac{\widehat{A}_i^{(j)}}{D_i^{(j)}} \left( \sum_{y=1}^{C} B_{iy}^{(j)} \log B_{iy}^{(j)} \right) - \frac{\widehat{B}_i^{(j)}}{D_i^{(j)}} \left( \sum_{a=1}^{C} A_{ia}^{(j)} \log \phi(Z_i^{(j)}, a) \right).
\end{aligned}
$$

For each $i \in [n]$, the computation of $\frac{\widehat{A}_i^{(j)}}{D_i^{(j)}} \sum_{y=1}^{C} B_{iy}^{(j)} \log p_y$, $\frac{1}{D_i^{(j)}} \left( \sum_{a=1}^{C} A_{ia}^{(j)} \log A_{ia}^{(j)} \right) \widehat{B}_i^{(j)}$, $\frac{\widehat{A}_i^{(j)}}{D_i^{(j)}} \left( \sum_{y=1}^{C} B_{iy}^{(j)} \log B_{iy}^{(j)} \right)$, and $\frac{\widehat{B}_i^{(j)}}{D_i^{(j)}} \left( \sum_{a=1}^{C} A_{ia}^{(j)} \log \phi(Z_i^{(j)}, a) \right)$ each takes $\Theta(C)$ time. As a result, the computation of $U_i^{(j)}$ for each $i \in [n]$ takes $\Theta(C)$ time. Therefore, the computation of $\frac{1}{n} \sum_{i=1}^{n} U_i^{(j)}$ takes $\Theta(nC)$ time.

Given $\phi(Z_i^{(j)}, a)$ for all $i \in [n]$ and $a \in [C]$, the conditional distribution matrix $Q^{(j)}(Z^{(j)} \in a \mid Z^{(j')} \in y) \in \mathbb{R}^{C \times C}$ can be efficiently computed following Algorithm 3. We denote by $Q^{(j)}[a, y] = Q^{(j)}(Z^{(j)} \in a \mid Z^{(j')} \in y)$ the $(a, y)$-th entry of $Q^{(j)}$, representing the conditional probability that a learned modality-specific representation $Z^{(j)}$ belongs to prototype $a$ given that the representation from the other modality $Z^{(j')}$ belongs to class $y$. Each entry $Q^{(j)}[a, y]$ is computed by aggregating the soft assignment values $\phi(Z_i^{(j)}, a)$ over all $i \in [n]$ such that $Y_i = y$, followed by normalization with respect to the total number of samples in that class. The accumulation step (lines 2–7 in Algorithm 3) requires $\Theta(nC)$ time, while the normalization step (lines 8–12 in Algorithm 3) requires $\Theta(C^2)$ time. Since $n \gg C$, the computational complexity of computing $Q^{(j)}$ is $\Theta(nC + C^2) = \Theta(nC)$.

Once $Q^{(j)}(Z^{(j)} \in a \mid Z^{(j')} \in y)$ is computed for $a \in [C]$ and $y \in [C]$, $V_i^{(j)}$ can be computed by $V_i^{(j)} = \sum_{a=1}^{C} \sum_{y=1}^{C} \phi \left( Z_i^{(j)} \in a, Z_i^{(j')} \in y \right) \log Q^{(j)}(Z^{(j)} \in a \mid Z^{(j')} \in y)$, which takes $\Theta(C^2)$ time for each $i \in [n]$ As a result, the computation of $\frac{1}{n} \sum_{i=1}^{n} V_i^{(j)}$ takes $\Theta(nC^2)$ time. Since $\text{IBB}^{(j)} = \frac{1}{n} \sum_{i=1}^{n} U_i^{(j)} - \frac{1}{n} \sum_{i=1}^{n} V_i^{(j)}$, the overall computation cost for $\text{IBB}^{(j)}$ is $\Theta(nT_0 + nC + nC + nC + nC^2) = \Theta(nT_0 + nC^2)$.

