# OpenReview forum: "IBMA: Information Bottleneck-Based Multimodal Alignment"
_ICML.cc/2026/Conference — ICML 2026 regular_

### Official Review · Reviewer_kLhU · 2026-03-03

**Soundness:** 3
**Presentation:** 2
**Significance:** 2
**Originality:** 2
**Overall Recommendation:** 4
**Confidence:** 3

**Summary:**

This paper proposes IBMA, which introduces a modality-specific Information Bottleneck constraint in addition to the fused representation, aiming to promote cross-modal alignment and suppress modality-specific noise. It also presents a variational upper bound (IBB) that does not rely on Gaussian assumptions. The method achieves consistent improvements on emotion recognition, sentiment analysis, and multimodal medical classification tasks.

**Compliance With Llm Reviewing Policy:**

Affirmed.

**Key Questions For Authors:**

What is the fundamental difference between the modality-specific IB loss and direct contrastive alignment (e.g., InfoNCE)?

Does IBB introduce estimation bias? Is there any theoretical error bound?

When the number of classes C is large (e.g., 1000-class tasks), does IBB still retain computational advantages?

Does the method provide stronger robustness under missing-modality or noisy-modality settings?

Could the modality-specific IB loss suppress modality-specific but useful information?

**Limitations:**

The method relies on class centroids, which may be unstable in fine-grained or long-tail settings.

The current formulation strongly depends on supervised labels and may not extend naturally to self-supervised multimodal pretraining.

Only classification tasks are evaluated; retrieval or generation tasks are not explored.

The computational complexity still scales with C², which may limit applicability in large-class problems.

More in-depth representation-level analysis is lacking.

**Strengths And Weaknesses:**

Strengths:

The paper correctly points out that existing IB-based multimodal methods (e.g., MIB, OMIB) apply the IB principle only to the fused representation, overlooking redundancy suppression and cross-modal alignment at the modality-specific level. This limitation is valid and well identified.

It treats cross-modal representations as supervision signals, replacing label-based retention terms in the modality-specific IB objective. Such a formulation is relatively uncommon within the IB framework and shows a certain degree of conceptual novelty.

The paper proposes a new variational upper bound (IBB), provides a complexity analysis, and compares it with CLUB. The theoretical development is relatively complete.

The experiments span datasets of varying scales, from small benchmarks to datasets with over 200K samples, resulting in reasonably comprehensive empirical validation.

Weaknesses:

Although modality-specific IB losses are introduced, the method fundamentally remains a combination of mutual information minimization/maximization, cross-modal consistency regularization, and centroid-based MI approximation. Overall, the framework can be viewed as an extension and engineering refinement of the IB paradigm rather than a new learning paradigm.

The claim of being “distribution-free” appears somewhat overstated. While the method avoids Gaussian assumptions, the MI estimation is still based on class-centroid soft assignments and effectively relies on proxy class probabilities as a plug-in estimator. This does not constitute a fully distribution-agnostic estimator, but rather avoids the VAE-style prior assumption.

The computational complexity comparison is somewhat selective. When comparing with CLUB, the paper does not discuss sampling-based approximations of CLUB, nor does it compare with other MI estimators such as MINE or InfoNCE.

---

> ### Author Rebuttal · Authors · 2026-03-31
>
> **Weaknesses**
>
> **1. “...extension and engineering refinement of the IB paradigm.”**
>
> We respectfully disagree that “the framework is merely an extension or engineering refinement of the IB paradigm”. IBMA introduces two key contributions: a novel modality-specific IB alignment method and a distribution-free upper bound for IB loss (see our response to Weakness 4, fC69 for details).
>
> **2. …proxy class probabilities as a plug-in estimator. **
>
> **The use of class-centroid soft assignment does not imply any distributional assumption”.
> Computing the class membership probabilities follows prototype-based modeling in prototypical learning (PL) [1], operating in continuous space without parametric assumptions. Our class-centroid estimators are not typical non-parametric plug-in estimators because the class centroids are learnable. Prior works [1] show that DNNs learn well-structured features aligned with class centroids, making such assignments reliable.
> In our framework, the information-theoretic objective (IBB) further encourages the learned features to align with class labels, thereby improving the quality of these probability estimates. Our class-centroid-based estimator can be extended to PL setup with stronger modeling capability (using prototypes as the class centroid) (See our response to Weakness 2 of reviewer HCMu.)
>
> **3. …sampling-based approximations of CLUB, …MINE or InfoNCE.**
>
> We compare IBB with sampling-based CLUB. As shown in the table below, while sampling reduces CLUB’s cost, it causes notable performance degradation, whereas IBB achieves significantly better results.
>
> |Methods|Training Time|Acc|
> |-|-|-|
> |CLUB (25% sampling)|2.3|63.1|
> |CLUB (50% sampling)|2.8|63.5|
> |CLUB|3.8|64.1|
> |IBB |2.0|65.4|
>
> MINE and InfoNCE are lower bounds of mutual information (MI) and are commonly used to increase MI between representations and labels. However, the IB principle aims to reduce IB loss by simultaneously increasing the MI between the learned representation and the target labels and reducing the MI between the learned representation and the input. As shown in the table below, IBMA significantly outperforms  MINE and InfoNCE.
>
> |Methods|MELD (%)|
> |-|-|
> |MINE|64.6|
> |InfoNCE|64.8|
> |IBMA|65.7|
>
> **Questions**
>
> **1. ”...direct contrastive alignment?”**
>
> See our response to Weakness 3.
>
> **2. “…theoretical error bound?”**
>
> Following PL [1] (our response to Weakness 2, HCMu), we extend class centroids to prototype/clustering centroids. With more prototypes, the centroid-based density better approximates the true continuous density, tightening the IB loss bound. The error corresponds to the gap between this estimate and the ground-truth density, which can be bounded in standard statistical analyses such as kernel density estimation [2].
>
> **3. “When the number of classes C is large…**
>
> As shown in our response to Weakness 3  of Reviewer 1WNj, IBMA outperforms competing methods on tasks with over 100 classes.
>
> **4. …noisy-modality settings?**
>
> Our IB loss is inherently robust to noise because reducing the IB loss (and also our IBB) reduces the mutual information between the representation and noisy inputs, making the learned features less correlated with noise and thus more robust.
>
> **5…modality-specific but useful information?**
>
> IBMA enforces the IB loss reduction on the fused representation, which decreases its mutual information with modality-specific inputs while increasing it with class labels to maintain task-relevant information.
>
> **Limitations**
>
> **1. “...long-tail settings.”**
>
> As shown in Table 4 and our response to Weakness 2 (HCMu), IBMA outperforms all methods on long-tail datasets MIMIC-CXR and CheXpert.
>
> **2. “...self-supervised pretraining.”**
>
> Since the prototype is computed by clustering [1], the IBMA-PL proposed in our response to Weakness 2 of reviewer HCMu can be applied to self-supervised pretraining (Linear Eval).
>
> |Method|UPMC|WIKI-DOC|
> |-|-|-|
> |CLIP-style Pretraining|88.7|90.8|
> |SimCLR-style Pretraining|88.1|90.2|
> |IBMA-PL (Self-Supervised)|90.4|92.3|
>
> **3. “...generation tasks.”**
>
> As discussed in Limitation 2, IBMA-PL can be extended to self-supervised multimodal pretraining, yielding general-purpose features for retrieval. Our work focuses on discriminative tasks, following [3].
>
> **4. “...large-class problems.”**
>
> As demonstrated in our responses to Weaknesses 3 of Reviewer 1WNj, IBMA significantly outperforms competing methods for tasks with more than 100 classes while maintaining a good training efficiency.
>
> **5. “Representation-level analysis.”**
>
> The t-SNE visualization in Link [4] shows that IBMA yields more compact, well-separated clusters.
>
> **References**
>
> [1] Caron et al. Unsupervised learning of visual features by contrasting cluster assignments. NeurIPS 2020
>
> [2] Kim et al. Robust kernel density estimation. JMLR 2012
>
> [3] Wu et al. Learning optimal multimodal information bottleneck representations. ICML 2025
>
> [4] https://anonymous.4open.science/status/IBMA-R

---

> > ### Author Rebuttal · Reviewer_kLhU · 2026-04-03
> >
> > My questions have been resolved, so I am keeping my score.

---

### Official Review · Reviewer_fC69 · 2026-03-04

**Soundness:** 2
**Presentation:** 2
**Significance:** 3
**Originality:** 2
**Overall Recommendation:** 4
**Confidence:** 4

**Summary:**

This paper proposes IBMA, a multimodal learning framework based on the Information Bottleneck principle. The method applies the IB objective not only to the fused multimodal representation but also to each modality-specific representation. It derives a new variational upper bound for the IB loss that does not rely on Gaussian assumptions and VAE latent space. The approach is evaluated on several multimodal tasks, including emotion recognition, sentiment analysis, and medical image–text classification, where it shows consistent improvements over prior methods.

**Compliance With Llm Reviewing Policy:**

Affirmed.

**Final Justification:**

Most of the issues have been satisfactorily resolved, particularly those related to the centroid-based formulation.
I increase my score to 4. However, I am not inclined to rate it higher, as I continue to have reservations regarding the originality of the work, which remain a significant concern. While I appreciate the authors’ additional comments on originality, I still view this work primarily as an extension of existing frameworks. This concern is also reflected in the assessments of other reviewers.

**Key Questions For Authors:**

* Can you better explain the theoretical motivation for the centroid-based construction and how it directly follows from the IB objective?
* Are the representations and centroids normalized during training? If not, how do you prevent trivial solutions related to scaling?
* How sensitive is the method to the choice of the balance parameter and to the number of classes when computing centroids?

**Limitations:**

yes

**Strengths And Weaknesses:**

* Soundness.
The objective is clearly defined, and the derivation of the variational upper bound appears correct and logically structured. The assumptions are reasonable, and the complexity analysis is clear. The empirical evaluation is extensive and covers different datasets and tasks. Comparisons with strong baselines are provided, and ablation studies support the role of modality-specific alignment.
However, some aspects need clarification. The centroid-based formulation is introduced quite abruptly (first appearance of word "centroids" in row 214L withouth context) and is not well motivated from the IB perspective. It is not fully clear how the centroid construction directly follows from the theoretical objective. Important implementation details are also missing, such as whether representations and centroids are normalized. Without such constraints, there could be trivial solutions where norms grow or collapse.

* Presentation. The paper is generally well structured, but the writing is dense and sometimes difficult to follow. The notation is heavy, which makes the derivations harder to read than necessary. The introduction of centroids and related probabilities would benefit from more intuition before presenting formulas.  The current illustration of the method feels more like a high-level sketch than a detailed figure, and it does not fully support the reader in understanding the key ideas behind the approach. In particular, when looking only at Figure 2, the model appears very similar to many existing multimodal architectures in the literature, namely two modality-specific encoders followed by a fusion module. I would kindly suggest further refining the figure to better highlight the distinctive components of the proposed method, especially the modality-specific alignment and the IB-based regularization.

* Significance. Applying the IB principle at both fused and modality-specific levels is a meaningful **extension** of prior work. The  derivation of a novel distribution-free variational upper bound for the IB loss (IBB) is noteworthy.
The consistent empirical gains across different domains, including medical tasks, suggest practical relevance.

* Originality. The idea of using the Information Bottleneck in multimodal learning is not new, but applying it explicitly to modality-specific representations in addition to the fused representation is a novel extension. The structured centroid-based formulation and the new variational upper bound add technical originality. The contribution is not a completely new paradigm, but it provides a thoughtful and well-justified development of existing ideas.

---

> ### Author Rebuttal · Authors · 2026-03-31
>
> **Responses to the Weaknesses and Questions**
>
> **W1. "The centroid-based formulation is introduced quite abruptly ... "**
>
> **Q1. “...motivation for the centroid-based construction…”**
>
> **Q2. “…centroids normalized during training…”**
>
> The introduction of the class centroids aims to calculate soft class-membership probabilities for the computation of the mutual information in the IB loss. The use of the class centroids and the computation of the probability that $Z_i^{(j)}$ belongs to class $a$ by $P[Z^{(j)} \in a] = \frac 1n \sum_{i=1}^n  \phi(Z_i^{(j)}, a)$ follows the prototype-based feature distribution modeling in existing prototypical learning methods [1,2,3,4], where extensive studies show that there is no need to normalize the centroids [1, 2].
>
> The training of IBMA starts with a 5-epoch warmup stage, when only the CE loss is optimized, to ensure that the latent features generated after the warmup stage are reasonable and informative. The centroids are computed as the average of the features after the warmup. In this way, the class centroids are already well and implicitly regularized as averaged features generated by a moderately trained neural network after the warmup, so there is no need for normalizing the centroids, and the norms of the centroids are free of the scaling problem in all the experiments in this paper. Such a strategy is also widely used in the literature [1, 2].
>
> **W2. "...kindly suggest further refining the figure to better highlight the distinctive components of the proposed method, especially the modality-specific alignment and the IB-based regularization."**
>
> We will add more technical details to Figure 2 so that readers can easily identify the novelties of IBMA different from the existing literature. We will refine  Figure 2 to better highlight the distinctive components of the proposed method, especially the modality-specific alignment and the IB-based regularization.
>
> **W4. "...The contribution is not a completely new paradigm..."**
>
> We respectfully and strongly disagree that our novelty is only “a thoughtful and well-justified development of existing ideas.”
> Our IBMA makes two-fold contributions: a novel modality-specific IB alignment method and a novel distribution-free upper bound for the IB loss.
>
> In contrast to existing IB-based methods that formulate multimodal IB objectives for fused representations, IBMA enforces the IB principle to both the fused representation and the individual modality-specific representations to learn informative and task-relevant representations with guidance from the other modality.
>
> IBB differs significantly from the IB upper bounds in the literature, such as VIB, APIB, and CLUB, as it is distribution-free and computationally efficient. VIB and APIB impose an unrealistic Gaussian assumption on the hidden features, and APIB reduces only an approximation to the IB loss. IBMA directly reduces a variational upper bound for the IB loss, IBB, without introducing any distributional assumptions or approximation to the IB loss. Moreover, the proposed IBB is significantly more efficient than CLUB, which does not require distributional assumptions.
>
> This is the first time that multimodal learning employs a new variational upper bound for IB and a new modality-specific IB alignment scheme, and our IBMA significantly outperforms the current state-of-the-art by a large margin, as demonstrated in Tables 1-6, owing to its novelties, which are also acknowledged by reviewer HCMu and kLhU.
>
>
> **Q3. ”How sensitive…the balance parameter and to the number of classes…”**
>
> The sensitivity of the performance of IBMA to the choice of the balance parameter has been studied in Section C.2 in the appendix of our paper. It is observed in Table 8 of our paper that IBMA maintains stable performance across different values of the balance parameter.
>
> When using the class centroids to compute the probability that a feature belongs to a class, the number of centroids is decided by the number of classes in the dataset. In our response to Weakness 2 of reviewer HCMu, we further extend the class-centroid formulation to prototypical learning (PL) [1,2], leading to IBMA-PL, where the number of centroids equals the number of prototypes, which is larger than the number of classes and is determined by standard cross-validation. As shown in the table below, the performance of IBMA-PL is stable with regard to the choice of the number of prototypes.
>
> |# Prototypes|MELD|
> |--|--|
> |$2\times C$|66.0|
> | $5\times C$  | 66.3 |
> |$10\times C$|66.3|
> |$15\times C$| 66.2 |
> |$20\times C$|66.2|
>
> **References**
>
> [1] Snell et al. Prototypical networks for few-shot learning. NeurIPS 2017
>
> [2] Yang et al. Robust classification with convolutional prototype learning. CVPR 2018
>
> [3] Caron et al. Unsupervised learning of visual features by contrasting cluster assignments. NeurIPS 2020
>
> [4] Li et al. Prototypical contrastive learning of unsupervised representations. ICLR 2021

---

> > ### Author Rebuttal · Reviewer_fC69 · 2026-04-03
> >
> > Thank you to the authors for taking the time to address my concerns. Most of the issues have been satisfactorily resolved, particularly those related to the centroid-based formulation. I would encourage the authors to further clarify and better motivate the use of centroids in the revised version of the paper.
> >
> > I am willing to increase my score to 4. However, I am not inclined to rate it higher, as I continue to have reservations regarding the originality of the work, which remain a significant concern. While I appreciate the authors’ additional comments on originality, I still view this work primarily as an extension of existing frameworks. This concern is also reflected in the assessments of other reviewers:
> >
> > 1WNj - "The proposed modality-specific alignment via Information Bottleneck shares significant conceptual commonalities with Multi-View IB and Conditional IB."
> >
> > HCMu - "This represents a reasonable extension of existing paradigms rather than a groundbreaking innovation that significantly shifts the research direction."
> >
> > kLhU - "Overall, the framework can be viewed as an extension and engineering refinement of the IB paradigm rather than a new learning paradigm."

---

> > > ### Author Response · Authors · 2026-04-06
> > >
> > > Thank you for your comments. Your concerns are further addressed below.
> > >
> > > We emphasize that the novelty and originality of our IBMA have been addressed in our response to reviewer 1WNj, HCMu, and kLhU. **Importantly, kLhU confirmed that the concerns were fully addressed, including the concern regarding the novelty of IBMA, indicating that IBMA is not solely an extension to existing methods**; reviewer HCMu did not mention the novelty issue anymore in the Rebuttal Acknowledgment. The “conceptual commonalities” in review 1WNj do not suggest any novelty issue. In fact, our IBMA and existing works, Multi-View IB and Conditional IB, all belong to the class of information-theoretic-based methods for multimodal learning, so that IBMA shares conceptual commonalities with existing IB-based methods.
> > >
> > > We emphasize the following key points in the novelty and significance of this work, which are more than solely an “extension” of existing IB-based multimodal learning methods.
> > >
> > > **Significant limitations in the literature addressed by our IBMA: IBMA makes existing non-applicable IB principles applicable to broad DNNs for multimodal learning.**
> > >
> > > **Requiring restrictive and unrealistic assumptions.** Representative methods, Multi-View IB [1] and Conditional IB [2], rely on an unrealistic distributional assumption about latent features of the DNN. Both Multi-View IB [1] and Conditional IB [2] rely on an unrealistic distributional assumption by adopting a VAE-style surrogate, wherein the conditional distribution $p(\zeta \mid z)$ is enforced to match the marginal $p(\zeta)$ assumed to be a fixed Gaussian prior. Herein $z$ denotes the learned task-relevant representation, and $\zeta$ is a latent feature introduced for multimodal representation learning, obtained by encoding $z$ through a VAE encoder.
> > >
> > > Such restrictive and unrealistic assumptions make the existing methods [1, 2] not applicable to realistic DNNs with complex distributions of the latent features. In a strong contrast, the IBMA framework is applicable to arbitrarily complex distributions of the latent features of DNNs due to the novel and distribution-free upper bound for the IB loss, the IBB, explaining its superior performance over the existing literature, as demonstrated in Tables 2-4 of our paper. **Such an improvement is fundamental rather than being solely an “extension”. Our IBMA renders non-applicable IB principles applicable to a broad class of DNNs for multimodal learning**.
> > >
> > > Furthermore, our IBB is computationally more efficient than the existing distribution-free upper bound for the IB loss [3]. The overall computational complexity for calculating the proposed modality-specific IBB regularization term $\textup{IBB}^{(j)}$ is $\Theta(nT_0 + nC^2)$, where $T_0$ denotes the computational complexity of a forward and backward pass through the neural network. In contrast, computing the upper bound for the mutual information required for calculating the distribution-free upper bound for the IB loss proposed in CLUB  [3] requires a substantially higher computational complexity, at least $\Theta(n^2 T_0)$ since $n \gg C^2$ on multimodal learning datasets detailed in Table 1 of our paper.  We note that $\Theta(n^2 T_0)$ corresponds exclusively to computing the upper bound for the mutual information $I(Z, X^{(j)})$, while CLUB additionally requires the computation of the lower bound for the mutual information $I(Z, Y)$. Details on the complexity analysis of CLUB and our IBB are presented in Section D of the supplementary. The advantages of IBB over existing upper bounds in terms of training time and performance are demonstrated in Table 5 of our paper.
> > >
> > > **Suggestion for Adhering to the ICML Reviewing Policy**. According to the ICML reviewing policies, our contributions are clearly distinguished from prior work, and our novelty is well justified. This reviewer also acknowledged the novelty of both the modality-specific alignment and the proposed upper bound (IBB), so we hope this reviewer will re-evaluate the significance of the novelty of IBMA, which is thoroughly explained above.
> > >
> > > Thank you very much for your time!
> > >
> > > **References**
> > >
> > > [1] Wu et al. Learning optimal multimodal information bottleneck representations. ICML 2025
> > >
> > > [2] Wang et al. Conditional Information Bottleneck for Multimodal Fusion: Overcoming Shortcut Learning in Sarcasm Detection. AAAI 2026
> > >
> > > [3] Cheng et al. Club: A contrastive log-ratio upper bound of mutual information. ICML 2020.

---

### Official Review · Reviewer_HCMu · 2026-03-11

**Soundness:** 3
**Presentation:** 2
**Significance:** 3
**Originality:** 3
**Overall Recommendation:** 3
**Confidence:** 4

**Summary:**

This paper proposes Information Bottleneck-Based Multimodal Alignment, a novel framework for multimodal representation learning. While existing methods predominantly apply the Information Bottleneck principle only to fused multimodal representations, this framework explicitly enforces constraints on individual modality-specific encoders. The authors derive a novel, distribution-free variational upper bound for the loss. This theoretical contribution avoids the restrictive fixed Gaussian prior assumptions common in conventional methods and enables highly efficient optimization via standard stochastic gradient descent. Extensive experiments demonstrate that the proposed method consistently outperforms existing state-of-the-art multimodal learning approaches.

**Compliance With Llm Reviewing Policy:**

Affirmed.

**Key Questions For Authors:**

1. The proposed bound relies heavily on class centroids, soft assignments, and label-aware statistics. Does this not introduce a strong structural assumption that the latent space must form easily separable, discrete clusters? How tightly does the method bound the true loss under conditions where such structural assumptions might fail, such as highly imbalanced data or overlapping class manifolds?
2.  Could the performance gains be attributed broadly to the addition of a cross-modal alignment objective rather than specific information-theoretic properties? How does the framework compare against simpler regularization baselines, such as direct feature consistency loss or cross-modal distillation applied at the modality-specific level?
3. While the proposed bound is computationally efficient, updating class centroids and statistics poses challenges for large-scale distributed training or datasets with massive vocabularies. Can the authors discuss or provide empirical evidence on scaling computationally to tasks with hundreds of classes? Furthermore, how does the pairwise alignment scale when the number of modalities increases beyond three?

I would be willing to raise my score if the authors could demonstrate the method's scalability to datasets with a large number of classes, and provide additional baseline experiments to clarify whether the performance gains genuinely stem from the sophisticated IBB formulation, or merely from the naive regularization of pulling image and text features closer together.

**Limitations:**

yes

**Strengths And Weaknesses:**

**Strengths:**
1. The authors accurately identify the pain point that existing methods ignore modality-specific noise, making the logic of introducing constraints early in the modality alignment highly sound. As an objective function-level enhancement, the approach does not bind to a specific backbone network and can be naturally integrated into standard architectures, making it highly reusable.
2. The experiments comprehensively span multiple highly diverse real-world scenarios, including emotion recognition, sentiment analysis, medical vision-language classification, and anomalous tissue detection. Testing across such varied domains convincingly demonstrates the generality of the proposed framework.
3. The paper provides complete ablation studies, computational complexity comparisons, sensitivity analyses, significance tests, and intuitive visualization maps, which multi-dimensionally support the core conclusions.

**Weaknesses:**

1. The core contribution lies in adding modality-specific alignment and a new upper bound on top of existing frameworks, while the main model remains a standard fusion architecture. This represents a reasonable extension of existing paradigms rather than a groundbreaking innovation that significantly shifts the research direction.
2. The derivation heavily relies on class centroids and label-aware statistics. This inherently implies a strong discrete clustering structure assumption, making its theoretical rigor in non-discrete tasks questionable. Furthermore, the paper lacks comparisons with simpler regularization methods or powerful pre-trained foundational models, making the exact source of the performance gains somewhat ambiguous.
3. Although the theoretical complexity is superior to existing methods, it still requires the global maintenance of class centroids. The actual computational overhead and engineering practicality remain inadequately discussed when facing extremely large class vocabularies, continuous regression tasks, distributed training, or pairwise alignment across more than three modalities.

---

> ### Author Rebuttal · Authors · 2026-03-31
>
> **Responses to Weaknesses (W) and Questions (Q)**
>
> **W1. "...reasonable extension of existing paradigms rather than a groundbreaking innovation..."**
>
> Our IBMA features two-fold contributions, including a novel modality-specific IB alignment method and a novel distribution-free upper bound for the IB loss. (see our response to Weakness 4 of fC69 for details).
>
> According to ICML reviewing policies, our contributions on the modality-specific alignment and the proposed upper bound (IBB), as acknowledged by the reviewer, are clearly distinguished from prior work, and our novelty is well justified. This work does not focus on neural architecture design, as it is outside our scope. Instead, our goal is to advance multimodal learning on top of existing architectures using the proposed novel methods.
>
> **W2 "... implies a strong discrete clustering structure assumption..."**
>
> **Q1 " ... How tightly does the method bound... highly imbalanced data..."**
>
> **Q2 ”...feature consistency loss or cross-modal distillation…"**
>
> The computation of the probability that a feature belongs to a class via aggregated soft assignments follows prototype-based feature distribution modeling in prototypical learning methods [1,2,3,4], which do not require a strong discrete clustering assumption on the learned features. **We emphasize that there is no need to make the “strong structural assumption that the latent space must form easily separable”. The success of the prototype-based methods [3,4] shows that neural networks can produce latent features that are well separated by class centroids. The information-theoretic objective, IBB, encourages learning latent features aligned with class centroids.
>
> To further improve the modeling capability over that of class centroids (Class-C), we extend our Class-C-based learning to prototypical learning (PL), leading to IBMA-PL, where we use prototypes (a.k.a the clustering centroids). The number of prototypes is larger than that of Class-C ($C$), which is selected by cross-validation  [1,2] from $C$ to $20 \times C$ with a step of $C$. The distribution of the latent feature space is better captured with more prototypes than classes [1,2].  The performance of PL is shown in the table below. We note that the computation of prototypes is efficient [1,2], as demonstrated in the table below.
>
> |Methods||MELD|MIMIC-CXR|CheXpert|
> |--|--|--:|--:|--:|
> |OMIB|Acc|64.3|71.0|89.3|
> ||Training Time|3.0|28.6|27.9|
> |IBMA|Acc|65.7|72.2|90.7|
> ||Training Time|3.2|30.2|29.4|
> |IBMA-PL|Acc|66.3|72.7|91.1|
> ||Training Time (Minutes/Epoch)|3.4|32.1|31.0|
>
> With more prototypes, the probability density modeled by PL is closer to the ground-truth continuous density, making the IB loss bound tighter. The theoretical error bound is the gap between centroid-based density estimation and the ground-truth density, which can be bounded in standard statistical analyses such as kernel density estimation [7].
>
> MELD, MIMIC-CXR, and CheXpert used in our paper are highly imbalanced datasets. For example, in MELD, the neutral class (4710) is 17.4$\times$ and 17.6$\times$ larger than disgust (271) and fear (268). The results in Tables 2 and 4, and the table above, show that IBMA significantly outperforms all competing methods. By introducing more cluster centroids via PL, IBMA-PL further improves performance on such highly imbalanced datasets.
>
> We compare IBMA with two ablation models that replace the modality-specific IB alignment loss with the CLIP-based contrastive loss [5] and distillation loss [6] for cross-modal feature alignment. As shown in the table below, IBMA significantly outperforms the two ablation models, highlighting its advantage over existing modality-specific regularization methods.
>
> |Methods|MIMIC-CXR|CheXpert|
> |--|--|--|
> |Contrastive Ablation Model|70.8|89.3|
> |Distillation Ablation Model|71.0|89.2|
> |IBMA|72.2|90.7|
>
> **W3 "...when facing extremely large class vocabularies..."**
>
> **Q3 "... hundreds of classes?...modalities increases beyond three?"**
>
> Class centroids are efficiently computed each epoch by aggregating features per class and averaging once. On CREMA-D, this adds only 2.1s per epoch (0.3% overhead, Sec. C.3). As shown in our responses to Weaknesses 3–4 of Reviewer 1WNj, IBMA outperforms competing methods on tasks with over 100 classes and 4 modalities while maintaining strong efficiency.
>
> **References**
>
> [1] Caron et al. Unsupervised learning of visual features by contrasting cluster assignments. NeurIPS 2020
>
> [2] Li et al. Prototypical contrastive learning of unsupervised representations. ICLR 2021
>
> [3] Yang et al. Robust classification with convolutional prototype learning. CVPR 2018
>
> [4] Snell et al. Prototypical networks for few-shot learning. NeurIPS 2017
>
> [5] Zhang et al. Multi-modal semantic understanding with contrastive cross-modal feature alignment. COLING 2024
>
> [6] Li et al. Decoupled multimodal distilling for emotion recognition. CVPR 2023
>
> [7] Kim et al. Robust kernel density estimation. JMLR 2012

---

> > ### Author Rebuttal · Reviewer_HCMu · 2026-04-02
> >
> > Thank you for the detailed rebuttal. I appreciate the new rebuttal results comparing against contrastive and distillation-style modality-level alignment baselines, which help reduce part of my earlier concern that the gains might come merely from adding a generic alignment regularizer. While the rebuttal argues that the method does not require a strong discrete clustering assumption, the practical formulation still appears heavily centered around class centroids, soft assignments, and label-conditioned statistics. The prototype-based extension discussed in the rebuttal is interesting and directionally helpful, but at present it reads more like a promising extension than a theoretical resolution within the paper itself. In particular, the statement that the approximation gap can be controlled by standard density-estimation arguments is currently too informal for me to view it as fully addressing the concern. The rebuttal suggests that the method can scale, but I would still prefer to see this reflected directly and more explicitly in the paper. Overall, the rebuttal improves my view of the empirical support for the method, especially with respect to the baseline ambiguity. However, it does not fully resolve my concerns about the generality of the theoretical formulation and the scalability claims.

---

> > > ### Author Response · Authors · 2026-04-02
> > >
> > > Thank you for your comments. Your concerns are further addressed below.
> > >
> > > **The accuracy (consistency) of the estimators for the mutual information in lines 184-187**
> > >
> > > To address your concern that “the statement that the approximation gap can be controlled by standard density-estimation arguments is currently too informal for me”, we provide more formal analysis with the consistency (that is, the approximation gap converging to $0$) between the prototype-based estimators and the corresponding true quantities. We analyze the estimator $\textup{Pr}[Z^{(j)} \in a] $ as an example, and the other estimators are analyzed similarly, such as $\textup{Pr}[Z^{(j)} \in a, X^{(j)} \in y] $,  $\textup{Pr}[Z^{(j)} \in a, Z^{(j’)}  \in y] $, and the estimators for the mutual information in line 184-187.
> > >
> > > The estimator $\textup{Pr}[Z^{(j)} \in a] = \frac{1}{n} \sum_{i=1}^{n} \phi(Z^{(j)}_i, a)$ is a consistent estimator. Let $\mathcal{F}^{(j)} = \lbrace F_1^{(j)}, \dots, F_C^{(j)} \rbrace$ be the set of learnable centroids (a.k.a. the prototypes), where $C$ is the number of prototypes. According to the theory of asymptotic quantization in (Pollard, 1982), for a fixed $C$, the centroids $\mathcal{F}^{(j)}$ optimized via empirical risk minimization converge almost surely to the optimal centers $\mathcal{F}^{(j)*}$ that minimize the expected quantization error:
> > >
> > > $$\min _ {\mathcal{F}^{(j)} } \frac{1}{n} \sum_{i=1}^{n}  \| Z^{(j)} _ i - F_ a^{(j)} \|^2   \overset{\textup{large } n }{\longrightarrow} \mathbb {E} \left[ \min _ { \mathcal{F}^{(j)}  }  \| Z^{(j)} _ i - F_a^{(j)}  \|^2 \right] \quad  (a). $$
> > >
> > > We note that the centroids $\mathcal F^{(j)} $ are computed as the averages of the features in each class, so $\mathcal F^{(j)} $ would minimize the LHS of Eq. (a), guaranteeing the consistency of $\mathcal F^{(j)}$.
> > >
> > > Furthermore, under the softmax formulation, $\phi(Z^{(j)}_i, F_a ^{(j)} )$ acts as a continuous relaxation of the hard assignment. The consistency of the estimator $\textup{Pr}[Z^{(j)}  \in a]$ then follows from the standard kernel density estimation literature (Einmahl, 2005; Goldstein, 1992; Rigollet, 2009), which establish consistency of the estimator $\textup{Pr}[Z^{(j)} \in a]$ together with the consistency result in Eq. (a) above.
> > >
> > > As a response to your comment that “I would still prefer to see this reflected directly and more explicitly in the paper”, **we will add all the results in this rebuttal, including the formal analysis above, to the final version of this paper, and this is the purpose of the rebuttal process**.
> > >
> > >
> > >
> > > **References**
> > >
> > > (Einmahl, 2005) Uniform in bandwidth consistency of kernel-type function estimators. Annals of Statistics
> > >
> > > (Goldstein, 1992) Optimal plug-in estimators for nonparametric functional estimation. Annals of Statistics
> > >
> > > (Rigollet, 2009) Optimal rates for plug-in estimators of density level sets. Bernoulli
> > >
> > > (Pollard, 1982) A central limit theorem for k-means clustering. Annals of Probability

---

### Official Review · Reviewer_1WNj · 2026-03-12

**Soundness:** 2
**Presentation:** 2
**Significance:** 2
**Originality:** 2
**Overall Recommendation:** 4
**Confidence:** 3

**Summary:**

This paper introduces Information Bottleneck-based Multimodal Alignment (IBMA), a framework designed to improve multimodal representation learning by applying the Information Bottleneck principle at two levels: the fused multimodal representation and individual modality-specific representations. IBMA utilizes a cross-modal supervision mechanism where representations from one modality guide the learning of another to suppress modality-specific noise. Experiments across emotion recognition, sentiment analysis, and medical disease classification datasets demonstrate that IBMA outperforms several state-of-the-art baselines.

**Compliance With Llm Reviewing Policy:**

Affirmed.

**Key Questions For Authors:**

- The notation $(Pr[Z^{(j)} \in a])$ treats the event of a representation “belonging to a class” as a probability, while $(Z^{(j)})$ is a continuous vector.

- In Algorithm 3, the soft assignments $(\phi(Z_i^{(j)}, a))$ are used as counts to update the distribution $(Q)$, but $(\phi)$ is a normalized softmax output.

**Limitations:**

Please see weakness.

**Strengths And Weaknesses:**

**Strengths**
- The paper extends the Information Bottleneck principle from fused multimodal representations to modality-specific representations with a clear architecture and theoretical motivation.

- The method is evaluated across several multimodal tasks and datasets, and supported by comparisons and ablation studies that suggest the proposed alignment mechanism can reduce IB loss and improve performance.

**Weaknesses**
- The proposed modality-specific alignment via Information Bottleneck shares significant conceptual commonalities with Multi-View IB and Conditional IB.

- The current ablation study focuses primarily on the removal of the modality-specific alignment term as a whole. It remains unclear whether the observed performance gains stem from the alignment strategy or the specific optimization properties of the IBB bound compared to VIB or CLUB.

- The sensitivity analysis only varies $\eta$, but the model's performance likely depends heavily on the number of classes $C$, which is small in all tested datasets ($C \le 7$). The scalability to large-label spaces is untested.

- Since IBMA is presented as a general multimodal framework, it would also be useful to examine how the method behaves under different numbers of modalities.

---

> ### Author Rebuttal · Authors · 2026-03-31
>
> **Responses to the Weaknesses**
>
> **1. “...commonalities with Multi-View IB and Conditional IB.”**
>
> Our novel upper bound for the IB loss, IBB, differs significantly from Multi-View IB [1] and Conditional IB [2], as it is distribution-free and computationally efficient. Multi-View IB [1] and Conditional IB [2] rely on an unrealistic distributional assumption. In contrast, IBB does not require distributional assumptions, leading to significantly better performance, as demonstrated in Tables 2-4. Furthermore, our proposed IBB is more computationally efficient than existing IB upper bounds that do not require distributional assumptions, as demonstrated in Table 5 of our paper.
>
> **2. "...gains stem from the alignment strategy or the … IBB..."**
>
> We conduct an ablation study in Section 4.6 by replacing IBB with existing upper bounds for IB, including VIB, APIB, and CLUB. As shown in Table 6, IBB significantly outperforms these existing upper bounds. For your convenience, we have performed ablation studies on them using the same datasets, MIMIC-CXR and CheXpert. The table below shows that both the Modality-Specific IB Alignment (MSIA) and IBB contribute to IBMA's superior performance. For instance, removing the modality-specific IB alignment or the IBB leads to 1.0% and 0.7% decreases on CheXpert.
>
> |Dataset|Methods|mAUC (%)|
> |-|-|-|
> |MIMIC-CXR|IBMA w/o MSIA|71.4|
> ||IBMA w/o IBB|71.7|
> ||IBMA|72.2|
> |CheXpert|IBMA w/o MSIA|89.7|
> ||IBMA w/o IBB|90.0|
> ||IBMA|90.7|
>
> **3. "...The scalability to large-label spaces is untested."**
>
> We perform additional experiments on two datasets with more classes $C$, UMPC Food-101 ($C$=101)  [4] and WIKI-DOC ($C$=111) [3]. We employ the pretrained ViT-B and BERT-base as image and text encoders. As shown in the table below, IBMA significantly outperforms all competing methods on both datasets, e.g., IBMA outperforms the best baseline, OMIB, by $1.2\%$ in accuracy on UPMC Food-101.
>
> |Method|UPMCFood-101 ($C$=101)|WIKI-DOC ($C$=111)|
> |-|-|-|
> |MMIB|90.2|92.1|
> |MMRLIB|90.3|92.3|
> |MIB|90.8|92.6|
> |MCIB|91.0|92.7|
> |CLFA|90.9|92.6|
> |DCLF|91.1|92.8|
> |OMIB|91.2|93.0|
> |**IBMA(Ours)**|**92.4**|**94.1**|
>
> |Dataset|OMIB Accuracy|IBMA Accuracy|OMIB Training Time (Minutes/Epoch)|IBMA Training Time (Minutes/Epoch)|
> |-|-|-|-|-|
> |UPMC Food-101 ($C$=101)|91.2|92.4|6.8|7.4|
> |WIKI-DOC ($C$=111)|93.0|94.1|7.6|8.1|
>
> **4. "...how the method behaves under different numbers of modalities."**
>
> We conduct additional experiments on PME4 [5], which contains four modalities, including audio, video, EEG, and EMG. We employ a pretrained CNN-based encoder for visual inputs, a pretrained audio encoder for speech signals, and lightweight neural encoders for EEG and EMG. As shown in the table below, IBMA significantly outperforms all competing methods on PME4, e.g., the best baseline, OMIB, by $1.1\%$.
>
> |Method|PME4|
> |-|-|
> |MMIB|79.8|
> |MMRLIB|80.0|
> |MIB|80.6|
> |MCIB|80.8|
> |CLFA|80.7|
> |DCLF|80.9|
> |OMIB|81.1|
> |**IBMA**|**82.2**|
>
> |Method|PME4 Accuracy|Training Time (Minutes/Epoch)|
> |-|-:|-:|
> |OMIB|81.1|4.3|
> |**IBMA (Ours)**|**82.2**|**4.6**|
> **Responses to the Questions**
>
> **1. "...``belonging to a class'' as a probability..."**
>
> The computation of the probability that $Z_i^{(j)}$ belongs to class $a$ by $P[Z^{(j)} \in a] = \frac 1n \sum_{i=1}^n  \phi(Z_i^{(j)}, a)$ follows the prototype-based feature distribution modeling in existing prototypical contrastive learning methods [6, 7]. Existing prototypical learning methods estimate class probabilities $P[Z^{(j)} \in a]$ by the same method with compelling performance.
>
> **2. "...$\phi(Z_i^{(j)}, a)$ are used as counts to update the distribution $Q$..."**
>
> Algorithm 3 computes $Q(Z_i^{(j)} \in a \mid Z_i^{(j’)} \in y)$ by summing $P(Z_i^{(j)} \in a)$ (normalized softmax outputs) over all data points in class $y$ for each $y$. $Q(a,y)$ is then divided by the number of data points in class $y$, which gives the variational probability $Q(Z_i^{(j)} \in a | Z_i^{(j’)} \in y) $. Using $\phi$ in $Q(a,y)$ produces normalized softmax (soft) probabilities instead of hard counts, improving robustness to noise and outliers, which is consistent with the reason probabilistic modeling (e.g., EM) that aggregates soft assignments as expected counts.
>
> **References**
>
> [1] Wu et al. Learning optimal multimodal information bottleneck representations. ICML 2025
>
> [2] Wang et al. Conditional Information Bottleneck for Multimodal Fusion: Overcoming Shortcut Learning in Sarcasm Detection. AAAI 2026
>
> [3] Fujinuma et al. A multi-modal multilingual benchmark for document image classification. EMNLP 2023
>
> [4] Wang et al. Recipe recognition with large multimodal food dataset. ICMEW 2015
>
> [5] Chen et al. Emotion recognition with audio, video, EEG, and EMG: a dataset and baseline approaches. IEEE Access 2022
>
> [6] Caron et al. Unsupervised learning of visual features by contrasting cluster assignments. NeurIPS 2020
>
> [7] Li et al. Prototypical contrastive learning of unsupervised representations. ICLR 2021

---

### Decision · Program_Chairs · 2026-04-30

**Decision:**

Accept (regular)

**Comment:**

The paper proposes a framework for multimodal alignment. It introduces modality-specific constraints along with a new variational upper bound for the IB loss. It has received four expert reviews. Three reviewers gave a Weak Accept recommendation, while one reviewer gave a Weak Reject. The authors provided a rebuttal, which addressed several concerns and led to a moderately positive overall consensus.

While the reviewers noted several strengths (i) a technically sound formulation with a clear extension of the IB principle to modality-specific representations (ii) a novel and computationally efficient variational upper bound for the IB loss and (iii) strong and comprehensive empirical evaluation across multiple datasets and tasks, they also raised some concerns regarding the incremental nature of the contribution, aspects of the theoretical formulation, and scalability considerations. Reviewer 2’s concerns related to generality and scalability needs special attention in the final version.

The paper has some useful ideas. The decision is to recommend acceptance.